# Heterogeneous effects of battery storage deployment strategies on decarbonization of provincial power systems in China

Liqun Peng [1], Denise L. Mauzerall [1,2] ✉, Yaofeng D. Zhong[3] & Gang He [4,5] ✉

Battery storage is critical for integrating variable renewable generation, yet how the location, scale, and timing of storage deployment affect system costs and carbon dioxide ($CO_2$) emissions is uncertain. We improve a power system model, SWITCH-China, to examine three nationally uniform battery deployment strategies (Renewable-connected, Grid-connected, and Demand-side) and a heterogeneous battery deployment strategy where each province is allowed to utilize any of the three battery strategies. Here, we find that the heterogeneous strategy always provides the lowest system costs among all four strategies, where provinces with abundant renewable resources dominantly adopt Renewable-connected batteries while those with limited renewables dominantly adopt Demand-side batteries. However, which strategy achieves the lowest $CO_2$ emissions depends on carbon prices. The Renewable-connected strategy achieves the lowest $CO_2$ emissions when carbon prices are relatively low, and the heterogeneous strategy results in the lowest $CO_2$ emissions only at extremely high carbon prices.

Carbon dioxide ($CO_2$) emissions from China's power sector reached ~5030 Tg in 2020[1], accounting for more than 40% of China's and 14% of global energy-related $CO_2$ emissions[1]. Decarbonizing China's power sector is essential for decarbonizing its economy and keeping the increase in global average temperature below 2 °C. In 2020, China pledged to peak its $CO_2$ emissions by 2030 and achieve carbon neutrality by 2060. Deep decarbonization of electricity generation is critical for achieving this goal, and the speed at which this decarbonization occurs will greatly influence China's total cumulative $CO_2$ emissions.

To achieve China's decarbonization goal while meeting increasing electricity demand, renewable energy must replace coal-fired power generation. A major challenge of increasing the penetration of wind and solar generation in the power system is their intermittency—their availability depends on the weather and time of day. Various technologies can smooth this variability, with energy storage being the most

promising[2–8]. Battery storage allows rapid energy discharges to smooth fluctuations in electricity supply. It also offers substantial storage capacity and can be deployed in various locations and strategies. Furthermore, the cost of battery storage has decreased rapidly in recent years, making it economically feasible for large-scale deployment. Thus, here we focus on batteries as the main source of energy storage for balancing variable renewable generation to achieve decarbonization goals.

The effects of battery storage on power systems have been explored in many countries[8–13], such as the US, EU, Australia, and India. While the benefits of battery storage are clear, deployment strategies involve complex energy, economic, and emission trade-offs. Some studies[14–17] highlight the importance of battery storage deployment strategies and their location in power systems. For example, Schmidt et al.[14] found that lifecycle greenhouse gas emissions and costs of storing electricity are determined by battery technology, applications,

[1]Princeton School of Public and International Affairs, Princeton University, Princeton, NJ 08544, USA. [2]Department of Civil and Environmental Engineering, Princeton University, Princeton, NJ 08544, USA. [3]Department of Mechanical and Aerospace Engineering, Princeton University, Princeton, NJ 08544, USA. [4]Department of Technology and Society, College of Engineering and Applied Sciences, Stony Brook University, Stony Brook, NY 11794, USA. [5]Marxe School of Public and International Affairs, Baruch College, City University of New York, New York, NY 10010, USA. ✉e-mail: mauzerall@princeton.edu; gang.he@baruch.cuny.edu

and geographies. Fares et al.[16] discovered that residential energy storage alone does not necessarily lead to reduced emissions if the stored energy comes from non-renewable sources like coal or natural gas. Hittinger et al.[17] illustrated that total $CO_2$ emissions are lower when storage is charged from collocated wind rather than directly from the grid. Craig et al.[18] showed that deploying battery storage would enable a shift of generation from gas- to coal-fired power and therefore increase $CO_2$ emissions of the power system in Texas. These findings underscore the need to consider deployment strategies and locations to enhance the cost-effectiveness and $CO_2$ mitigation associated with battery storage in power systems.

Other studies[19–22] focus on the role of battery storage deployment in China's power system. For example, He et al.[5] and Liu et al.'s[22] research suggests that the deployment of energy storage systems can help reduce carbon emissions by facilitating renewable energy integration and improving the overall efficiency of the power system. However, the effects of different storage deployment strategies on total system costs and $CO_2$ emissions at the national/provincial level in China and at multi-year scales are underexplored in this literature.

China's government has encouraged various battery storage deployment strategies. Since 2021, local governments and power grid enterprises put forward "centralized renewable energy + energy storage" development incentive policies[1,23,24]. These batteries are referred to as 'Renewable-connected' (RE-connected) batteries since they are charged only from co-located renewables (wind or solar). China's government also set a goal of increasing 'Grid-connected' and 'Demand-side' battery storage to achieve a flexible and robust grid system. Grid-connected batteries are the most flexible type of storage. Electricity used to charge Grid-connected storage could be from either fossil or non-fossil fuels, such as wind, solar and nuclear power, and the stored energy can be discharged back to the central grid. Demand-side battery storage is deployed in local power grids and connected to demand loads, such as industrial parks or commercial buildings, usually to act as an emergency energy source and to achieve power arbitrage. For instance, electric vehicles could provide mobile battery storage and could connect to commercial buildings or residential apartments as Demand-side batteries. Due to safety concerns, the local distribution power grid cannot transmit power back to the central power grid in China; consequently, energy stored in demand-side batteries can only meet local demand.

To understand how different types of battery storage strategies affect power system decarbonization, our research first explores the effects of battery deployment strategies on China's power system costs and $CO_2$ emissions. We begin with three nationally uniform strategies, two carbon prices and three battery cost trajectories (See Table 1, "Scenario design" in "Methods"; and Fig. 1). For each scenario, our method chooses the installed power, battery capacity, power generation and interprovincial transmission that minimizes system costs through an improved capacity expansion power system model (Switch-China) (See "Methods"). These results provide policymakers with the cost-minimizing location, timing, and scale of uniform renewable generators and battery storage installation across China up to 2050. To further understand the effect of various battery storage deployment strategies on different provinces, we also examine a "Mixed" option that allows each province to install any of the three types of battery storage—RE-connected, Grid-connected, and Demand-side batteries (Table 1). We combine the mixed strategy with both low and high carbon prices as well as a rapid battery cost decrease to set up two additional scenarios—Mixed-Low-R and Mixed-High-R. We then compare the system cost and $CO_2$ emissions of each cost-minimizing solution at national and provincial levels across all scenarios.

Our results show that RE-connected batteries decrease $CO_2$ emissions from coal-fired power generation in provinces with abundant renewable resources by up to 9% compared with Grid-connected batteries. However, RE-connected batteries increase $CO_2$ emissions

from coal-fired power generation by up to 69% in provinces with limited renewable resources compared with Grid-connected batteries. Demand-side and RE-connected batteries have opposite effects on provincial coal-fired power generation and national transmission when each is compared with Grid-connected batteries. Battery deployment strategies have a direct impact on national electricity transmission capacities and costs. RE-connected batteries require the lowest transmission capacity among the three strategies, which results in the lowest transmission costs. The Mixed battery strategy results in the lowest total system costs compared to any nationally uniform battery deployment strategy. It can also achieve the lowest national $CO_2$ emissions when the carbon price is at least 66 times higher than the low carbon price scenario examined in our study. Our results identify that renewable resources, availability of transmission, and characteristics of demand determine the system costs and $CO_2$ emissions of heterogeneous provincial battery storage deployment strategies.

## Results
### Trade-offs between national carbon emissions and power system costs
In general, trade-offs between carbon emissions and electricity costs occur in the three nationally uniform battery storage deployment strategies. With both uniform low carbon prices and rapid battery storage cost decreases across the three deployment strategies, $CO_2$ emissions decrease rapidly from 2025 to 2050, with electricity costs decreasing rapidly from 2035 to 2050 due to the decreasing renewable generation costs during this period (Fig. 2). Comparing these three strategies, the RE-connected battery strategy achieves the lowest cumulative $CO_2$ emissions from China's power system from 2025 to 2050 (Supplementary Tables 7 and 8). This is because, under the RE-connected strategy, batteries can only be charged from renewable power generation, resulting in more solar and wind power and less coal-fired power generation being built to meet demand, which results in the least total $CO_2$ emissions of any uniform strategy. Electricity costs of the RE-and Grid-connected strategy remain similar, while the Demand-side battery strategy costs are much higher. In 2050, system costs of the Demand-side battery strategy are ~13% higher than Grid-connected or RE-connected battery strategies (Fig. 2). The total carbon emissions and power system costs are shown in Supplementary Tables 5 and 6. We also find that high carbon prices and rapid decrease of battery storage costs play key roles in large-scale battery deployment and rapid decarbonization of the power system (See Supplementary Note 3 and 4, Supplementary Figs. 9 and 10).

### National electricity transmission costs
Large-scale deployment of renewables will entail greater electricity transmission capacity. However, depending on the chosen battery deployment strategy, necessary transmission capacity and costs will vary. At the national level between 2025 and 2050, we find RE-connected batteries reduce the most needed transmission capacity of the three battery deployment strategies (Fig. 3a). This is because locally generated renewable power meets a larger portion of the local demand compared with other strategies so that less power is transmitted. We also find that peak transmitted power of the RE-connected battery strategy is reduced the most (Fig. 3b), resulting in the lowest transmission costs (Fig. 3d).

The Demand-side battery strategy requires the highest transmission capacity (Fig. 3a), which results in the highest transmission costs (Fig. 3d). However, this strategy actually transmits less electricity than the Grid-connected battery scenario in 2045 and 2050 (Fig. 3c). This can be explained by Fig. 3b—transmission power profiles of the three strategies in an example two-day period in 2050. The peak of the transmission power profile indicates transmission capacity requirements, where the Demand-side strategy is slightly larger than other strategies. The areas under the curves represent transmitted electricity, where the

**Table 1 | Scenario designs—National uniform and provincial heterogenous battery deployment strategies**

| Scenarios | Battery storage deployment strategies | Carbon prices | Battery storage cost reduction trajectories |
|---|---|---|---|
| RE-Low-B | Batteries charged only with co-located RE in each province (RE-connected batteries) | Low carbon price | Base case |
| RE-Low-M | | | Moderate battery cost decreases |
| RE-Low-R | | | Rapid battery cost decreases |
| RE-High-B | | High carbon price | Base case |
| RE-High-M | | | Moderate battery cost decrease |
| RE-High-R | | | Rapid battery cost decrease |
| Grid-Low-B | Batteries charged by and discharged to grid (Grid-connected batteries) | Low carbon price | Base case |
| Grid-Low-M | | | Moderate battery cost decrease |
| Grid-Low-R | | | Rapid battery cost decrease |
| Grid-High-B | | High carbon price | Base case |
| Grid-High-M | | | Moderate battery cost decrease |
| Grid-High-R | | | Rapid battery cost decrease |
| Demand-Low-B | Batteries charged and discharged on demand side in each province (Demand-side batteries) | Low carbon price | Base case |
| Demand-Low-M | | | Moderate battery cost decrease |
| Demand-Low-R | | | Rapid battery cost decrease |
| Demand-High-B | | High carbon price | Base case |
| Demand-High-M | | | Moderate battery cost decrease |
| Demand-High-R | | | Rapid battery cost decrease |
| Mixed-Low-R | A mix of RE-, Grid-connected, and Demand-side batteries | Low carbon price | Rapid battery cost decrease |
| Mixed-High-R | A mix of RE-, Grid-connected, and Demand-side batteries | High carbon price | Rapid battery cost decrease |

The results from RE-Low-R, Grid-Low-R, Demand-Low-R and Mixed-Low-R are presented in the main text. The results from other scenarios are presented in Supplementary Notes 3 and 4, Supplementary Figs. 9, 10, 15–22 and Supplementary Tables 7 and 8.

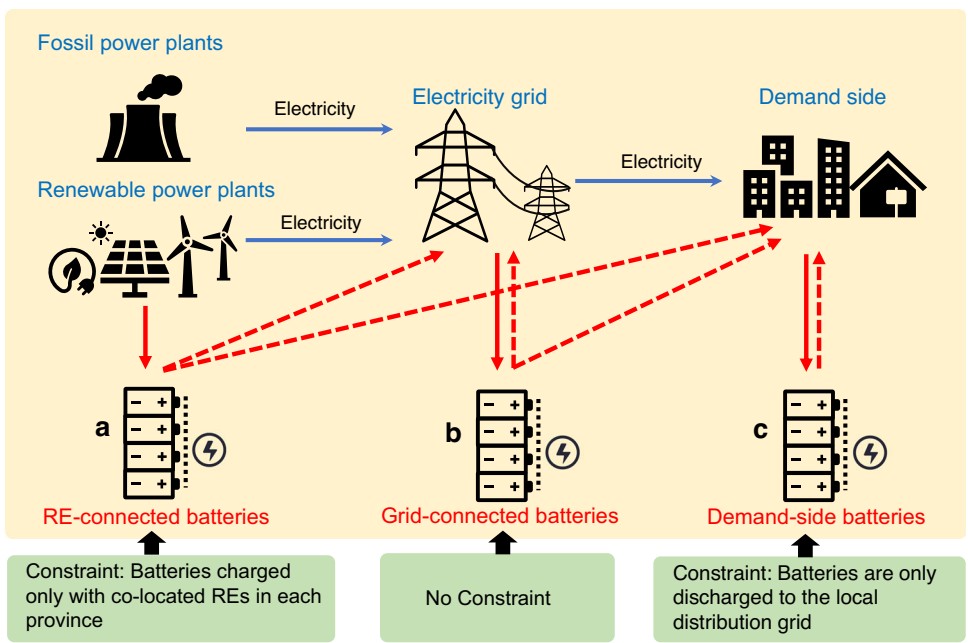

**Fig. 1 | Three battery storage deployment strategies that can be deployed either uniformly nationally or individually within each province. a** Renewable-connected (RE-connected) batteries: batteries that are co-located and only charged with RE; **b** Grid-connected batteries: batteries attached to the central grid with no additional constraints; and **c** Demand-side batteries: batteries co-located with demand loads and discharging only to the local grid. Solid red arrows show the electricity used to charge batteries. Dashed red arrows show electricity discharged from batteries.

Demand-side strategy is less than the Grid-connected battery, even though its peak transmission is higher. This explains the changing magnitudes of the three strategies in 2040 and 2045–2050 in Fig. 3a, c. We also conduct a sensitivity analysis about the effects of various technology costs on installed capacity, transmitted power, and annual transmission costs in Supplementary Figs. 20–22. We find that a more rapid decrease in technology costs results in larger differences among the three national battery strategies.

### Effects of nationally uniform battery deployment strategies
Our results show that nationally uniform battery deployment strategies have varying effects on individual provinces. We use the

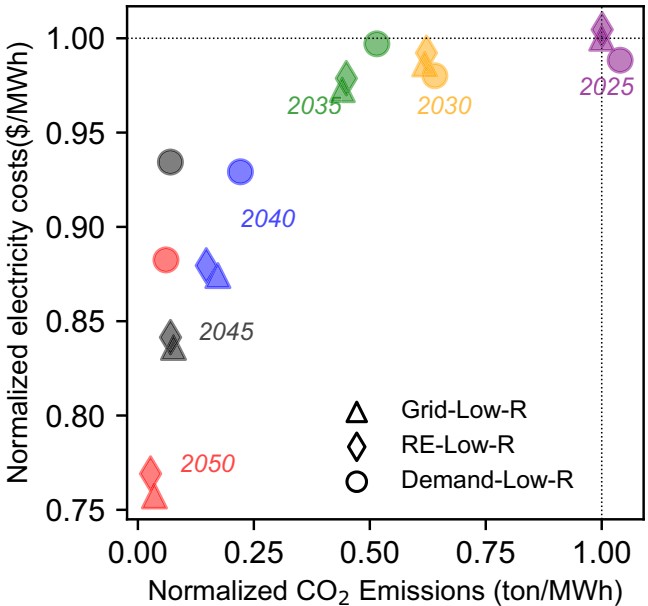

**Fig. 2 | National average optimal CO₂ emissions and optimal electricity costs per MWh of three uniform battery deployment strategies from 2025 to 2050.** Different colors represent different years. The electricity cost of each strategy is calculated as the annual system costs divided by the annual generated electricity. Grid-Low-R, RE-low-R and Demand-Low-R are RE-connected, Grid-connected and Demand-side with low carbon prices and rapid decrease of battery costs. Source data are provided as a Source Data file.

Grid-connected strategy as the baseline scenario since battery storage in this strategy can be flexibly used on either the supply or demand side. We find that the effects of the RE-connected and Demand-side battery strategies mainly depend on the characteristics of provincial energy generation and demand. We observe that certain provinces' total local power generation exceeds their total demand; we refer to these provinces as supply provinces. Their optimal local coal-fired power generation decreases under the RE-connected battery strategy and increases under the Demand-side battery strategy relative to the Grid-connected strategy. On the other hand, we refer to provinces where local power generation is less than total demand as demand provinces. Their optimal local coal-fired power generation increases under the RE-connected battery strategy and decreases with the deployment of a Demand-side battery strategy. Supplementary Fig. 12 shows the distribution of supply and demand provinces under most situations. We explain these provincial characteristics below using the energy balance equation.

$$\mathcal{D} = \mathcal{DE} + (\mathcal{B}_d - \mathcal{B}_c) + \mathcal{T}_{net} \qquad (1)$$

The equation shows the energy balance of each province, which is the key to explaining the effects of battery deployment strategies on power generation and CO₂ emissions. Here, $\mathcal{D}$ represents the provincial demand, which equals the sum of three terms—locally dispatched electricity $\mathcal{DE}$, local net battery discharge (the difference between battery discharge $\mathcal{B}_d$ and battery charge $\mathcal{B}_c$), as well as the net transmitted electricity into the province $\mathcal{T}_{net}$. The dispatched electricity $\mathcal{DE}$ equals generated electricity minus curtailed electricity. Curtailment ratios of renewables are shown in Supplementary Fig. 11. Equation 1 shows that the sum of these three terms must balance the demand of the province at each time step from 2025 to 2050 as well as

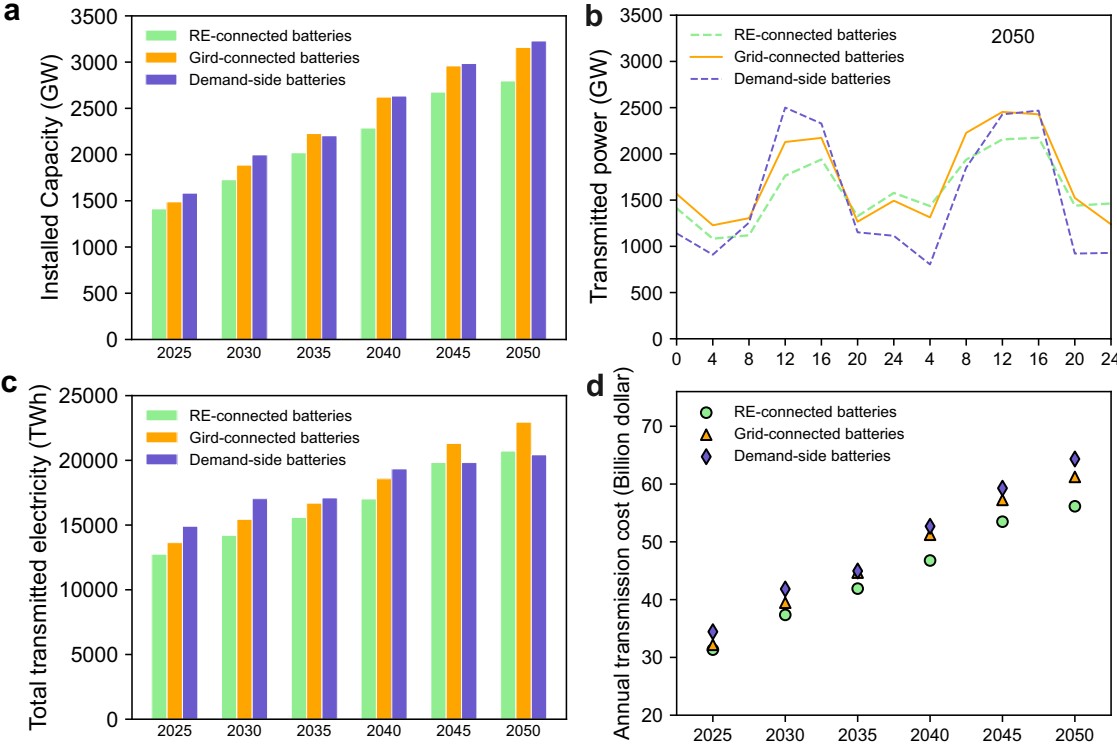

**Fig. 3 | Transmitted power and transmission costs of RE-connected, Grid-connected, and Demand-side battery deployment strategies over time with low carbon prices and rapid battery costs decreases.** Comparison of **a** optimal transmission capacity, **b** a comparison of optimal transmission power profiles for

the three deployment strategies for 2 days in 2050, **c** optimal total transmitted electricity and **d** optimal annual transmission costs for the three battery deployment strategies. Source data are provided as a Source Data file.

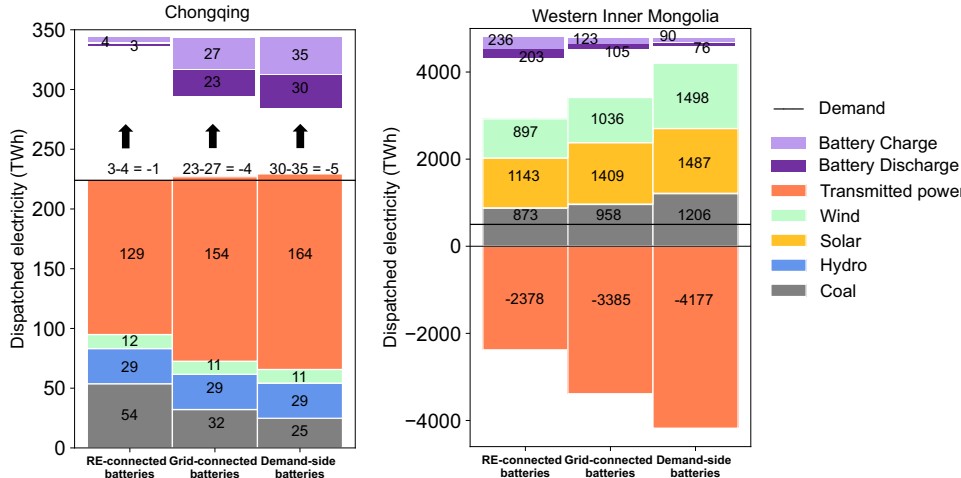

**Fig. 4 | Energy balance of Chongqing province and western Inner Mongolia in 2040 using nationally uniform RE-connected, Grid-connected, and Demand-side batteries with low carbon prices and accelerated battery cost decreases.** Chongqing province is a typical demand province, which has relatively high local demand and low local electricity supply. Western Inner Mongolia is a typical supply region, which has relatively low local demand and high coal and renewable supply. Positive (Negative) values indicate local production or transmission into (out of) the province. Black lines indicate provincial-level electricity demand; colored bar segments indicate the source of electricity supplied and transmitted electricity into/out of the province. For Chongqing and Western Inner Mongolia, numbers at the top of each column indicate battery charge/discharge, with the difference between the two resulting from efficiency losses. Source data are provided as a Source Data file.

annually. Analyzing the annual energy balance provides insight into the effects of battery deployment strategies.

We choose Chongqing and western Inner Mongolia as typical demand and supply provinces, respectively, in 2040, a time of considerable difference between these two provinces. Fig. 4 shows the annual dispatched power of Chongqing and western Inner Mongolia. In 2040, the electricity demand of Chongqing is more than twice its power generation. The lack of power generation is fulfilled by power transmitted in from other provinces. The power generation of western Inner Mongolia is around 6–8 times its electricity demand. The excess power generation is transmitted out to other provinces via transmission lines. As the demand of a province remains the same regardless of battery deployment strategies and power balancing must be satisfied, the strategies can only affect power generation, battery charge and discharge, and net transmitted power in Eq. (1).

The Grid-connected strategy results in supply province batteries being charged by power generated in the supply province. Depending on the province, this power generation could be dominated by either fossil or non-fossil generation. In a demand province, Grid-connected batteries will be charged with a power mix that is a combination of power generated locally and power transmitted in from other provinces.

Unlike the Grid-connected strategy, the RE-connected battery strategy includes the constraint that batteries can only be charged by local renewable generators. In demand provinces, this constraint leads to a reduction in installed batteries as well as a reduction in net transmitted power into the provinces, as they cannot rely on transmitted power to charge their batteries. Thus, we observe that in Chongqing province, battery discharge in the year 2040 decreases from 23 TWh in the Grid-connected strategy to 3 TWh in the RE-connected strategy, and net transmitted power decreases from 154 TWh to 129 TWh under the RE-connected strategy (Fig. 4). To balance the demand, local power generation increases from 72 TWh to 95 TWh in 2040. As coal is the cheapest and most available option in Chongqing, coal-fired power generation increases from 32 TWh to 54 TWh, accounting for 96% of the power generation increase. As demand provinces require less transmitted power from supply provinces, transmission from supply provinces decreases. In fact, we observe that while western Inner Mongolia supplies 3385 TWh in the

Grid-connected battery strategy, it only supplies 2378 TWh in the RE-connected battery strategy. In order to achieve power balance in western Inner Mongolia, fewer batteries and less power generation are needed with RE-connected rather than grid-connected batteries.

The effect of the Demand-side battery strategy on supply and demand provinces is opposite to that of the RE-connected battery strategy. In the Demand-side strategy, batteries continue to be charged from the grid, but we add a demand-side constraint that battery storage can only be discharged to local distribution power grids. As supply provinces cannot discharge battery power to the central grid in this strategy, there is a decrease in their optimal battery storage. In fact, we observe a decrease in battery discharge from 105 TWh to 76 TWh in western Inner Mongolia as Demand-side batteries displace Grid-connected batteries (Fig. 4). This decreased battery discharge has two effects. First, RE needs to be transmitted immediately to demand provinces rather than stored in batteries, which increases the need for additional electricity transmission to serve peak generation periods. Second, during periods of low generation in supply provinces and high demand in demand provinces, both RE power generation and battery discharge decrease, so more fossil fuel power generation is required to balance the demand. Therefore, we observe an increase in coal power generation in western Inner Mongolia from 958 TWh to 1206 TWh when Demand-side batteries displace Grid-connected batteries. As for demand provinces like Chongqing, as more power is transmitted from supply provinces (from 154 TWh to 164 TWh), more batteries are needed to store this energy, and thus the local power generation decreases (from 72 TWh to 65 TWh).

Figure 5 shows the annual coal-fired power generation and power discharged from battery storage from 2025 to 2050. Over each time step analyzed, compared with the Grid-connected battery strategy, the RE-connected battery strategy decreases coal power generation in the supply province (western Inner Mongolia) and increases it in the demand province (Chongqing), while the Demand-side battery strategy has the opposite effect. The Grid-connected battery strategy typically falls between the RE-connected and Demand-side battery strategies. Our results also indicate that as carbon prices increase over time, coal-fired generation decreases and battery storage increases, as expected (Supplementary Table 7). Dispatched power and load

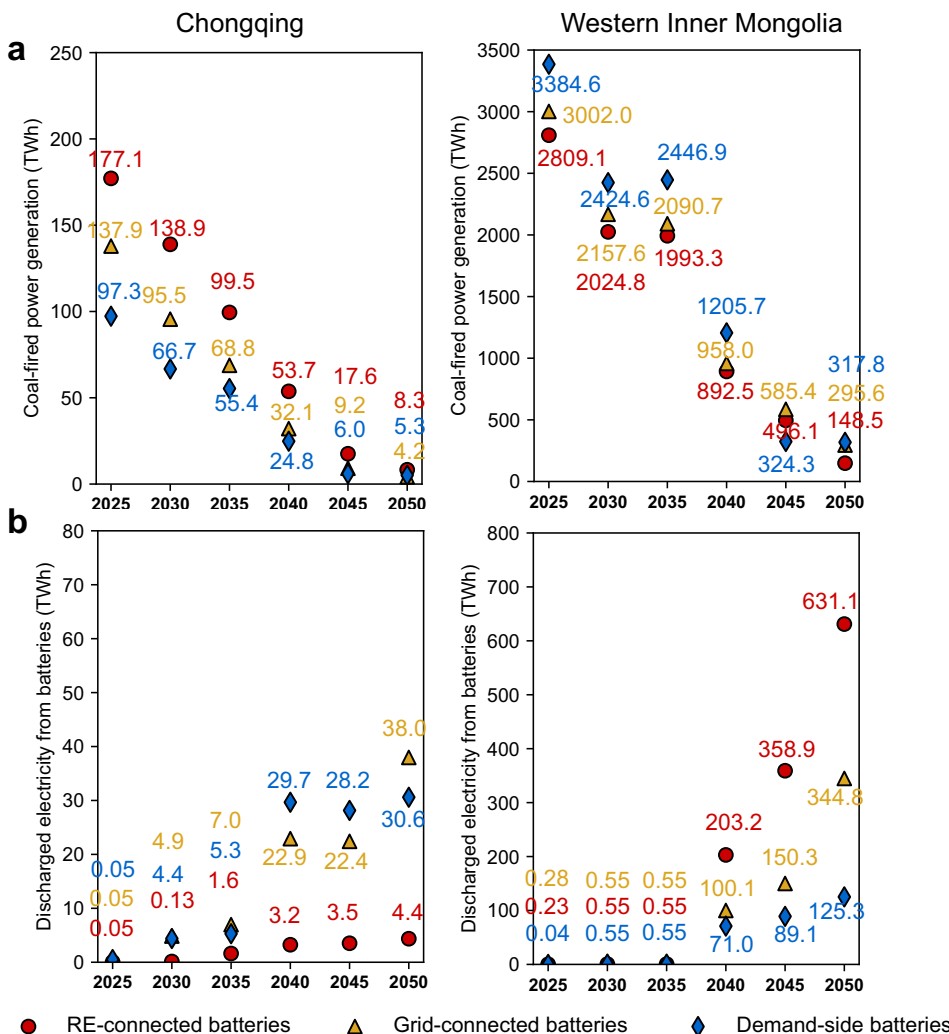

**Fig. 5 | Coal-fired power generation and power discharged from batteries in a typical supply and demand province.** Comparison of **a** coal-fired power generation, **b** power discharged from batteries in a supply province (western Inner Mongolia) and a demand province (Chongqing) for the three uniform battery deployment strategies with low carbon prices and rapid battery cost decreases. Note the very different scales for Chongqing and Western Inner Mongolia, which occur because Chongqing is a typical province with power demand that exceeds supply with little renewable resources, while western Inner Mongolia is a province with a demand far smaller than its power supply and a large supply of renewable resources. Effects of other scenarios are shown in Supplementary Figs. 15–18. Numbers in red, yellow, and blue represent the actual value of coal-fired power generation and discharged electricity from batteries under RE-connected, Grid-connected, and Demand-side battery strategies. Source data are provided as a Source Data file.

profiles from 2025 to 2050 for Chongqing and western Inner Mongolia under three battery deployment strategies are shown in Supplementary Figs. 13 and 14.

### Provincial heterogeneous battery deployment strategies

Besides uniform battery deployment strategies, we propose an additional strategy, referred to as the Mixed battery deployment strategy, where each province may utilize any of the three types of battery deployment. The Mixed battery strategy has the lowest total costs among all considered strategies. This is because, unlike the uniform strategies, the Mixed strategy does not constrain all provinces to employ only one type of RE-connected, Grid-connected, or Demand-side batteries. As this is a minimization problem with the total cost as the objective, the optimized solution of the Mixed strategy cannot have a total cost greater than any of the uniform battery strategies.

The Mixed strategy reconfirms our previous finding that the lowest-cost battery deployment strategy in each province depends on provincial characteristics—energy generation, demand, and renewable resources. By allowing each province to select a battery deployment

strategy to minimize national costs, we find that depending on the province's net power demand, the optimal battery deployment will vary (Fig. 6 and Supplementary Fig. 19). We find that Chongqing, a typical demand province, installs 100% Demand-side batteries in both Mixed-Low-R and Mixed-High-R. This is consistent with our findings on battery deployment choices under a uniform national strategy (Fig. 4). In the nationally uniform strategies Chongqing installs more batteries under the Demand-side deployment strategy than under the Grid- or RE-connected strategies. This is because, for demand provinces, their local power generation is less than the total demand, and they usually have limited renewable resources. Demand-side batteries can store electricity transmitted from supply provinces during the day and meet the high demand during the evening. In western Inner Mongolia, a typical supply province, under the Mixed scenario, most installed batteries are RE-connected—62% in Mixed-Low-R and 79% in Mixed-High-R. In both scenarios, only 3% of the installed batteries are Demand-side batteries. In the nationally uniform strategies, western Inner Mongolia installs more batteries under the RE-connected deployment strategy than in the other two nationally uniform

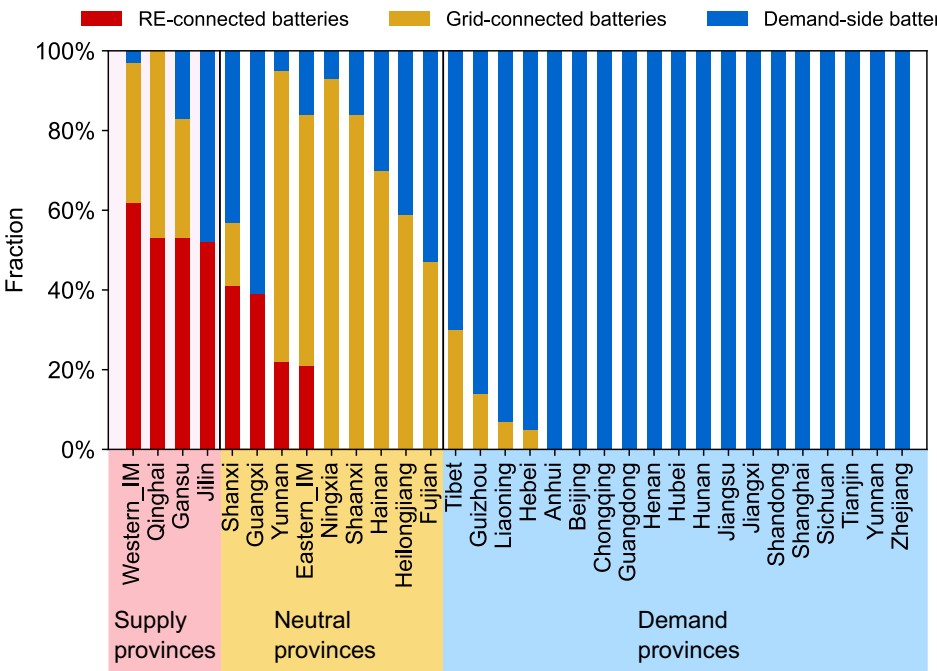

**Fig. 6 | Share of discharged power from each type of battery in the Mixed-Low-R scenario (see Table 1b for scenario descriptions).** Results for the Mixed-High-R scenario are shown in Supplementary Fig. 19. Mixed-Low-R and Mixed-High-R represent Mixed battery strategy with low (high) carbon price and rapid decrease of battery costs. Provinces in the pink shade are supply provinces, those in the yellow shade are neutral provinces, and those in the blue shade are demand provinces. Western_IM and eastern_IM represent western Inner Mongolia and eastern Inner Mongolia. Source data are provided as a Source Data file.

strategies. The reason is, for supply provinces, their total local power generation exceeds their total demand through 2050, and these supply provinces are mostly in western China with low population density and abundant renewable resources. RE-connected batteries in supply provinces can store excess renewable electricity and transmit it to demand provinces to meet their electricity needs. Therefore, in the mixed strategy, supply provinces choose to install more RE-connected batteries than Demand-side batteries, while demand provinces make the opposite choice.

The cumulative $CO_2$ emission in the Mixed battery strategy is slightly higher than RE- and Grid-connected battery strategy due to the adoption of demand-side batteries (Supplementary Tables 7 and 8). However, the differences in cumulative $CO_2$ emission between the Mixed battery strategy and the RE-connected battery strategy (the lowest $CO_2$ emissions strategy) decrease with the increase of carbon prices. When carbon prices become high enough (66 times greater than the low carbon prices in our study), cumulative $CO_2$ emission in the Mixed battery strategy will be the lowest among all battery strategies (See details in Supplementary Note 6).

## Discussion

Batteries will play a key role in the rapid decarbonization of the power system. With the increasing penetration of renewables, batteries can balance short-term electricity supply and demand while providing load shifting to fill gaps during peak demand and improving grid stability and providing consistent current. The rapidly decreasing costs of batteries provide opportunities for their rapid scale-up. However, how to best deploy battery storage at the sub-national level remains an open question.

We first examine the effects of nationally uniform battery storage deployment strategies and then heterogeneous deployment strategies (allowing each province to deploy a mixture of any of the three battery strategies it chooses) with varying battery costs and carbon prices on provincial carbon emissions and power system costs in China. We find that the Mixed strategy in intra- and inter-provincial battery

deployment results in the lowest system costs. Supply provinces with abundant renewable resources can minimize costs by having more RE-connected batteries and fewer Demand-side batteries, while demand provinces with limited renewable resources can minimize costs by deploying more Demand-side batteries. Although the cumulative $CO_2$ emissions in the Mixed battery strategy are not the lowest, the gaps of $CO_2$ emissions become smaller between the Mixed battery strategy and the RE-connected strategy with the growth of carbon prices. Our model results show that the Mixed battery strategy can achieve both the lowest total costs and $CO_2$ emission when carbon prices increase to 66 times that of the low carbon prices. Also, a rapid decrease in battery storage costs will significantly increase the scale of battery deployment and correspondingly reduce $CO_2$ emissions. High carbon prices will accelerate decarbonization levels while only marginally increasing system costs. With high carbon prices, carbon neutrality of the power sector can be achieved by 2050 if battery costs decrease moderately to rapidly. Our findings provide valuable insights into optimal battery deployment policies for China that facilitate carbon neutrality in the power sector. Our work is the first, of which we are aware, that analyzes battery storage options for each province individually before optimizing for minimum cost at the national level. Our disaggregated work (that allows differentiation between provinces rich in coal and rich in renewable generation) not only confirms early national findings but also provides a higher resolution for the variability existing among provinces.

Our results indicate that the effects of nationally uniform battery deployment strategies on national electricity transmission and provincial coal-fired power generation vary by province. Compared to nationally uniform Grid-connected batteries, we find that optimally deployed RE-connected batteries effectively reduce peak and total electricity transmission, with annual transmission costs reduced by ~9%. Also, coal-fired power generation decreases by ~9% in provinces with abundant renewable resources. This occurs because provinces with abundant renewable energy using renewable connected batteries transmit less locally generated electricity to other provinces than they

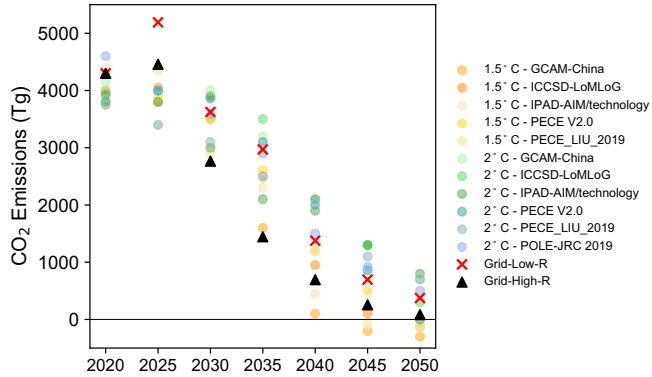

**Fig. 7 | Comparison of China's power sector $CO_2$ emissions in our study (Grid-Low-R and Grid-High-R) with other modeling studies[25].** 1.5 °C and 2 °C indicate $CO_2$ emissions consistent with an increase in the global average temperature of 1.5 °C and 2 °C target, respectively. Grid-Low-R and Grid-High-R is Grid-connected battery strategy with low (high) carbon prices and rapid decrease in battery costs. Source data are provided as a Source Data file.

would under the grid-connected scenario. Less transmission leads to less local coal power generation because less local power generation is required. However, coal-fired power generation is increased by up to 69% in provinces with limited renewable resources. This is because these provinces rely on electricity transmitted from other provinces, and less transmission requires more local power generation to balance the demand. As renewable resources are limited and coal-fired electricity is the cheapest alternative, an increase in coal-fired power generation is observed in these high-demand provinces.

When nationally uniform battery deployment takes place, Demand-side and RE-connected batteries have opposite effects on provincial coal-fired power generation and national transmission when they are compared with Grid-connected batteries (Figs. 2 and 5). Coal-fired power generation is increased by up to 26% in provinces with abundant renewable resources that use Demand-side batteries. This is because fewer batteries are installed as they are connected to the local grid and cannot discharge to the central grid. With fewer batteries to smooth the intermittency of renewable generation, more coal-fired electricity is required to meet local demand. However, coal-fired power generation is decreased by up to 22% in provinces with limited renewable resources since more power is transmitted into these provinces and less local power generation is needed.

When each province is free to choose its battery deployment strategy (Mixed strategy which allows each province to install any of the three types of battery storage: RE-connected, Grid-connected, and Demand-side batteries), we find that the Mixed strategy in intra- and inter-provincial battery deployment results in the lowest system costs. To minimize costs, demand provinces with limited renewable resources install far more Demand-side than Grid-connected or RE-connected batteries while supply provinces with abundant renewable resources primarily install RE-connected batteries.

In this study, we use carbon price as a driver to accelerate the decarbonization of China's power system. In order to verify the rationality of our carbon price assumptions, we compare the $CO_2$ emissions from our two scenarios Grid-Low-R and Grid-High-R, with other modeling results of $CO_2$ emissions in China's power system[25]. Fig. 7 shows that the $CO_2$ emissions in our study are within the ranges of other studies, except for higher $CO_2$ emissions in 2025 under Grid-Low-R. Unlike the other studies, we assume $CO_2$ emissions will continue to increase until 2025 before decreasing. Our assumption is based on President Xi's announcement at the Leader Summit on Climate, 2021, that China will strictly limit the increase in coal consumption over the 14th 5-year plan period (2021–2025) and only

begin to phase down coal power projects in the 15th 5-year plan period (2026–2030). We therefore assume that coal-fired power generation will continue to increase from 2020 to 2025 before decreasing after 2025.

We find that strategic deployment of battery storage will influence both system costs and carbon emissions from China's power system. To further accelerate the transition to a carbon-neutral power system, targeted government programs that provide financial incentives to encourage the adoption of RE-connected batteries in supply provinces and Demand-side connected batteries in demand provinces would facilitate more rapid power system decarbonization. In addition, improving transmission capacity and efficiency to facilitate the exchange of renewable energy between supply and demand provinces could maximize the benefits of battery deployment strategies.

Carbon pricing is a useful tool to guide investment decisions, which helps clean energy technologies compete with carbon-intensive alternatives and encourage more efficient energy use. Developing important decarbonization technologies, such as carbon capture and storage, batteries, and hydrogen storage, relies on long-term policies, such as carbon price signals. Our model does not include the potential effects of carbon prices on long-term renewable and battery costs. Although our modeling results identify possible pathways to accelerate the decarbonization of China's power sector under the eighteen scenarios developed for this study, government policies, stakeholder interests, and capital market constraints, among other factors, will influence our results in the real world. We use Li-ion batteries to represent all batteries in our study. However, as battery storage technology advances, alternatives should be assessed and incorporated into future battery storage deployment policies. In addition, batteries can offer ancillary services at a lower cost than traditional sources, such as gas-fired peaker plants. By participating in markets for ancillary services, batteries can generate revenue and offset their costs, making them a cost-effective option for providing these services. In future research, deploying a unit commitment power system model would be helpful for exploring these benefits further.

## Methods
### SWITCH-China model
We extend the SWITCH-China capacity expansion model[26] and use the extended model to conduct experiments. SWITCH-China is an optimization model which solves for the optimal installed capacities of generators and transmission line capacities to meet a specific demand. The optimal solution minimizes the cost of producing and delivering electricity while satisfying a set of operational constraints. In other words, given the electricity demand projection at a certain temporal and spatial resolution, SWITCH-China optimizes the number of each type of generator, energy storage, and transmission lines built each year in order to balance the projected demand with supply within each time step in each load zone[27,28]. The electricity demand projection at this resolution is detailed in Supplementary Note 1. Demand profiles in each investment period correspond to 2015, for which historical data are available for hourly loads, simulated hourly wind and solar capacity factors, and monthly hydroelectric availability. Supplementary note 5 shows the Sensitivity analysis of demand loads on battery deployment. The cost information for renewable power generators and fossil fuel power generation is shown in Supplementary Note 2 and Supplementary Table 4.

In our study, we use 1 h as a time step and a province as a load zone. To model power system dynamics, four levels of temporal resolution are employed by the SWITCH-China model: 5-year investment periods, months, days, and hours. Our study divides the time span 2025–2050 into six 5-year long investment periods: 2023–2028, 2028–2033, 2033–2038, 2038–2043, 2043–2048, and 2048–2053. We use 12 months to characterize each investment period, 1 day to characterize each month (the median load day) and 6 h to characterize

each day. For each day, hourly sampling begins at midnight China Standard Time (CST) and includes the 0th, 4th, 8th, 12th, 16th, and 20th hours. This results in (6 investment periods) × (12 months/ investment period) × (1 day/month) × (6 h/day) = 432 study hours over which the system is dispatched. The results of 2025, 2030, 2035, 2040, 2045 and 2050 are representative years for the six investment periods and are shown in our figures.

## Objective function

The default objective function in SWITCH-China is the national system cost. In other words, SWITCH-China outputs a power system plan that minimizes system cost by default. In this study, we model the power system to minimize total costs and explore the effect of various carbon prices on $CO_2$ emissions. Thus, we set up the objective function in SWITCH-China to be a weighted sum of system costs and $CO_2$ emissions.

$$\text{Minimize} : \mathcal{C} + \sum_{i \in \not{i}} \omega_i \mathcal{E}_i \qquad (2)$$

Here $\mathcal{C}$ represents the total system costs, $\mathcal{E}_i$ represents the total $CO_2$ emissions in period i, and the weight $\omega_i$ is the carbon price in period i. The carbon price captures the trade-off between cost efficiency and environmental impact, as increasing guides the model to output a system with lower $CO_2$ emissions at a higher system cost, and vice versa. The carbon prices used in our study are detailed in Supplementary Table 9.

The national system costs in the objective function include (1) capital costs of new power plants, battery storage and transmission lines, (2) operation and maintenance (O&M) costs of existing and new power plants, battery storage and transmission lines, and (3) variable costs incurred by power plants, e.g., fuel costs of coal and gas. Variable costs of wind, solar, pumped-hydro storage, power-gas-power (PGP) and battery storage are assumed to be zero. In general, these costs depend on both installed capacities within each province, which are model variables to be solved, as well as the unit capital costs and O&M costs, which are model inputs. The unit costs of wind and solar are obtained from the annual technology baseline (ATB) 2021[29] and IRENA, 2021[30], and detailed in Supplementary Note 2. The capital costs of battery storage are detailed in the following section. The capital costs of PGP storage technologies are evaluated from the model data compiled by the National Renewable Energy Laboratory (NREL)[31–33]. As for coal, natural gas, oil, hydro and nuclear power plants, their unit costs are derived from He et al.[5].

$CO_2$ emissions in the objective function come from burning coal, natural gas, and oil in the process of power generation. We calculate $CO_2$ emissions using energy consumption based on input $CO_2$ emission factors and the total power generation from coal and gas power generators (variables to be solved for in the optimization). In our study, the $CO_2$ emission factors are derived from the China Provincial GHG Emission Inventory Guideline (in Chinese) released by China's National Development and Reform Commission (NRDC). For fossil fuel power plants, we use average values for emission factors for coal-fired power plants of 25.41 kgC/GJ (93.2 $kgCO_2$/GJ), for natural gas power plants of 15.32 kgC/GJ (56.2 $kgCO_2$/GJ), and for oil power plants of 21.1 kgC/GJ (77.4 $kgCO_2$/GJ).

## Scenario design

In this study, we consider three factors—battery deployment strategies, carbon prices, and battery storage cost trajectories. In particular, we examine three battery deployment strategies (RE, Grid, and Demand), two carbon prices (Low, High), and three rates of battery storage cost decreases (Base, Moderate, Rapid). We then combine these three factors to set up eighteen scenarios (Table 1). The motivation and implementation of each factor are explained below.

## Battery deployment strategy

Three primary types of battery deployment are considered—RE-connected (RE), Grid-connected (Grid), and Demand-side (Demand), as well as a mixed battery deployment option that includes RE, Grid, and Demand batteries (Mixed). Grid-connected batteries are the most flexible since they can be charged from any type of generator. In the SWITCH-China model, we attach grid-connected batteries to the central grid with no additional constraints. RE-connected batteries are co-located with renewables (solar photovoltaic and wind) in order to smooth intermittent renewable generation before transmitting it to the central grid. Since 2021, some provinces in China have required that new renewable power plants be equipped with energy storage devices to smooth intermittency before power is transmitted to the grid. This is exactly the RE-connected battery scenarios considered in this study. In addition to attaching RE-connected batteries to the central grid, we add a constraint in the SWITCH-China model for them—in each province and at each time step, the total electricity used to charge these batteries does not exceed the power generated by the renewables. This constraint does not prevent renewables from being transmitted to other provinces and does not prevent discharged battery electricity from being transmitted to other provinces. Demand-side batteries are attached to the local power grid and are co-located with end users such as industrial factories or commercial buildings. In provinces that implement peak and valley electricity prices, the Demand-side battery strategy could help users reduce electricity bills and achieve peak-to-valley arbitrage. Also, in addition to fixed battery storage, electric vehicles could provide mobile battery storage. Due to existing infrastructure limitations in China, local power grids cannot transmit power back to the central power grid. Thus, in the SWITCH-China model, we attach Demand-side batteries to the local grid, which allows them to be charged from the central grid but only discharged to the local grid. For Mixed battery deployment, we assume that each province can use any mix of battery deployment strategies that minimizes both national and provincial costs. Regardless of the deployment strategy, we consider three battery durations—1, 4, and 10 h—to represent various types of battery durations. The lifespan of batteries is 15 years. In all our scenarios, we use a roundtrip energy loss from charging/discharging of 15% for 4-h batteries derived from NREL, 2021[29]. For 1-h and 10-h batteries, we assume roundtrip loss of 12% and 20%, respectively.

## Carbon prices

Faster decarbonization leads to lower cumulative $CO_2$ emissions, potentially at a higher cost due to earlier expansion of renewable generation and storage before their prices have decreased. He et al.[5] showed that high carbon prices can significantly increase storage capacity and reduce $CO_2$ emissions, thus achieving rapid decarbonization. In our model, we assume that the decarbonization rates of the power system are driven by carbon prices. This makes our results relevant to policymakers since the role of carbon pricing in climate policies is widely acknowledged[34–36]. We set up (1) low carbon prices, which allows the power sector to slowly transition away from fossil fuels to low carbon power generation. The low carbon price scenario indicates a moderate level of policy stringency, allowing the power sector to gradually shift from fossil fuels to low carbon power generation. and (2) high carbon prices, which enables a fast transition to low carbon power generation, reducing $CO_2$ emissions by 98% in 2050 compared to 2025 (Supplementary Table 9). The high carbon price scenario represents a high level of policy stringency and a strong commitment to achieving carbon neutrality in the power system by around 2050. Low carbon price from 2025 to 2035 is obtained from the projection by the State Grid Energy Research Institution (China Energy & Electricity Outlook 2019); the low carbon price after 2035 is based on our own assumptions that the power system will not achieve carbon neutrality by 2050 if there are not stringent carbon mitigation policies. High carbon price is set twice the value of low carbon prices from 2025 to 2050 to represent a more aggressive carbon emission mitigation target.

**Battery storage cost reduction trajectories.** We assume the cost of battery storage will decrease over time as technology advances. Chen et al.[20] pointed out that reductions in storage costs could effectively decrease the overall cost of carbon abatement and impact the scale of storage deployment. High uncertainties exist in current battery storage cost projections for the next few decades as the cost might decrease faster than expected[4]. Thus, we propose three battery cost trajectories to gain insight into how battery cost trajectories affect optimal battery storage deployment. The base trajectory (B) is constructed from the battery storage capital costs in NREL's ATB 2021[29] and storage futures study (2021). For the moderate trajectory (M), we assume projected battery costs are one-third of those in the base trajectory. For the rapid trajectory (R), we assume the projected battery costs are one-fifth of those in the base case. The cost trajectories for all types of batteries are illustrated in Supplementary Fig. 2.

## Data availability

The source data underlying Figs. 2–7 are provided as a Source Data file. The input data for the model are provided at GitHub: https://github. com/switch-model/switch-china-open-model/tree/main/inputs. Other data used for this analysis are available from cited publicly available sources or from the authors upon reasonable request. Source data are provided with this paper.

## Code availability

The key codes of SWITCH-China are provided at GitHub: https:// github.com/switch-model. Other code generated to analyze results and create figures are provided at GitHub: https://github.com/switch-model/switch-china-open-model.

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

## Acknowledgements

This work was funded by the Princeton School of Public and International Affairs (D.L.M. and L.P.) and the Princeton International and Regional Studies program (L.P.). G.H. was funded by the Global Energy Initiative at ClimateWorks Foundation (no. 23–2515). We thank Matthias Fripp for the instructions and insights on SWITCH model updates. We thank Nicolas Choquette-Levy for constructive comments on our manuscript.

## Author contributions

L.P. and D.L.M. conceived the idea of this project and designed the research. G.H. provided the data necessary to run the SWITCH-China model. L.P. conducted the optimization experiments and analyzed the results. Y.D.Z. and G.H. contributed to the SWITCH-China model updates. L.P. and D.L.M. wrote the manuscript. L.P., Y.D.Z., G.H. and D.L.M. reviewed and approved the final manuscript.

## Competing interests

The authors declare no competing interests.
