## [Peer review file · Nature Communications]

REVIEWER COMMENTS

Reviewer #1 (Remarks to the Author):

This work considers how battery deployment policies/strategies in China can affect grid emissions and transmission needs. The unique contribution is in the nature of the results: by asking a clear modeling question, the authors have addressed an important policy/regulation topic. In my view, the results and framing of the work are well-done and interesting. I do have several concerns and questions about the modeling approach and how it relates to those conclusions, some of which seem important to clearly resolve.

General Comments:

Overall, in these three general comments, I am concerned because I don't understand what is driving the different choices that the model is making. The paper is focused on the outcomes of the model (emissions, transmission builds, etc), but I don't understand why the model is making the choices that it is making – in many cases, I would expect it to make different choices.

1. In section 2 (and several other places): You frequently point out that the RE-sited battery strategy implies the lowest transmission requirement, but I expect that there is an important unstated corollary to this. As far as I can tell, the grid-sited batteries should provide the most benefits to the system, as the RE- and demand-sited scenarios only add constraints to the grid-sited batteries. That is to say: RE- and demand-sited batteries should be strictly inferior from a “battery value” point of view – the grid-sited batteries can do everything that either RE- or demand-sited batteries can do (and more). This seems to be what is described in the methods section. And if that is true, then grid-sited batteries should always be preferable from the point of view of the overall optimization: regardless of the nature of the scenario or objective (lowest cost, lowest emissions, some combination, etc), grid-sited batteries should always be better at achieving it. So while the RE- or demand-sited constraints may reduce emissions or transmission needs, those are (I suspect) always in the context of grid-sited being the most valuable way to use batteries. So RE-sited batteries may reduce transmission needs more than grid-sited, that is because the additional transmission is of limited cost and decreases the overall system costs in the grid-sited situation. Another way to say it: if limiting transmission was the goal, grid-sited batteries should be no worse than RE-sited batteries (and possibly better) at achieving it.

Overall, I may be misunderstanding the modeling approach or assumptions, but if it is the case that grid-sited batteries are always the best at achieving the objective function (it seems like they should be), then this is a very important thing to discuss, even though it would reduce the implied importance of studying the effects of alternative strategies.

2. Another modeling question, possibly related to the one above: if RE- and demand-sited storage are just constrained versions of the grid-sited deployment, then grid-sited should always be preferable (it is strictly more capable). But then why would the “Mixed” strategy be chosen at all? If grid-sited batteries can do everything that RE- and demand-sited ones can do, I would expect that the model should choose 100% grid-sited if offered the option under a “Mixed” strategy. In line 310, you do a good job of explaining why a “mixed” strategy is strictly better than a homogenous strategy – it can do everything that a homogenous strategy can do and more. I would think that this same argument would apply to grid-sited storage relative to the other two types. Line 307 says “The Mixed battery deployment strategy, in which each province may utilize a mixture of the three deployment strategies, has the lowest total costs of all considered strategies.” Why would this be better than 100% grid-sited, if grid-sited batteries provide the most flexibility?

3. This wasn’t clearly stated, but I get the impression from the wording that the optimization is done at a province level – is that right? That is, you aren’t optimizing for the lowest-cost system for all of China, but rather each province is trying to minimize its internal cost. Whichever way the optimization is designed, it shouldn’t affect my comments in #1 and #2 above. But if things are optimized at the province level, how do the provinces choose a Mixed strategy? Wouldn’t the optimal strategy in a province depend on the strategy used by the neighbors – a game theory problem? Given that the main arguments in the work are about how different battery strategies encourage or discourage export of electricity, it seems very relevant to ask what the neighbors are planning, especially if I expect to import or export a lot of power.

Minor Comments:

-I don’t think this recent paper is a problem for the work here, but I do think it may make a good comparison point for this analysis, and as a reader of both papers I’m curious if you feel that the results are in agreement or (if not) why the results differ: Mingquan Li, Rui Shan, Edgar Virguez, Dalia Patiño-Echeverri, Shuo Gao, Haichao Ma, Energy storage reduces costs and emissions even without large penetration of renewable energy: The case of China Southern Power Grid, Energy Policy, Volume 161, 2022, 112711, ISSN 0301-4215, <https://doi.org/10.1016/j.enpol.2021.112711>.

Reviewer #2 (Remarks to the Author):

This paper uses a power sector capacity expansion model to investigate how the location, scale, and timing of battery storage deployment alters its benefits for various decarbonisation scenarios in China. Although there is a large literature looking at regional storage benefits and value, the paper extends the literature by providing clear scenarios and figures to illustrate alternate locational strategies for batteries across Chinese provinces. The topic and results would be of interest to a range of stakeholders in light of debates about the role of battery storage and decarbonisation strategies in China.

There are a few issues the authors should revisit to improve the analysis and manuscript. Overall, the manuscript is well written, but there is uneven coverage of scenarios, with some places having too much explanation and others too little. In my view, a missed opportunity in the results section is that the authors do not use figures or text in the body to describe how these alternate strategies alter the generation mix, capacity mix, and configurations of storage. Supplementary Figure 5 does a nice job of showing many of these metrics, and I'd find it valuable to have this in the main text with an accompanying figure that shows variation in capacity by technology (perhaps focusing on a couple particular time periods to show scenario variation). This would help readers to understand implications of these strategies for investments.

There were several instances where additional information on the analysis framework and assumptions are important for situating the reader and helping them to better understand the analysis and its potential limitations without necessarily digging through the supplemental material or related papers. SWITCH is a well-established open-source model of capacity expansion, but several areas could be highlighted, in some instances mentioned in the body instead of the Methods section:

- Mention temporal and spatial resolution up front.
- Indicate stringency of power sector policies in terms of levels of price in 2050 and percentage CO₂ reductions. Or something similar to place the stringency of these policies in context.
- Mention choice set of storage options up front. And also that battery duration seems endogenous but only in pre-specified levels of 1, 4, and 10 hours.
- Figure 1: Is storage in the 'RE-connected batteries' case connected to solar and wind only? What are the assumptions about cost reductions from coupling? Is ILR endogenous?
- Lines 171-175: How are carbon prices included in the costs?
- Demand profile source? How much electrification is embedded in these shapes? The authors should consider a sensitivity on load shapes given how previous work indicated that these shapes materially impact storage deployment.
- At the beginning of Section 2.4, I initially missed that these are not only mixed strategies across provinces but also mixed within a province, too. It would be good to make this extra clear.

Another issue is that the literature review is rather limited. The contributions of this article relate to its geographical coverage and detailed investigation of the value of storage relative to where it is located. However, it would be valuable for readers to see the logic of this contribution expanded while also referencing other prominent studies in the US, EU, and others.

I had a number of additional suggestions as I reviewed the paper:

- Line 96: Considering using 'cost-minimizing location, timing, and scale' instead of 'best location...'
There are many dimensions to locational choice, so 'best' seems too broad and potentially normative.
- Figure 1: For Demand-side batteries, I could see where a reader might think these are V2G batteries from EVs based on the picture. Perhaps worth clarifying this in the text and image.
- Line 138: 'Requires' is too strong, perhaps 'entails'. And do you mean 'Large-scale deployment of renewables'? All else equal, storage and transmission are substitutes for each other, as your results demonstrate.
- Figure 2: It would be good to mention scenarios in first sentence of caption instead of at the end. I expected a little more discussion in this section of possible changes in these metrics across the carbon price and storage cost scenarios.
- Lines 167-168: It would be helpful to have a more detailed explanation for why this occurs. I think I understand, but it would be valuable for readers to have this spelled out.
- Lines 173-175: Can this be phrased more directly and quantitatively? Perhaps, 'Moving from low to high carbon prices increases battery storage from Y to Z GW in 2050, and lower costs increases deployment from Y to Z...'
- Lines 224-228: How are system losses accounted for here? I didn't find this equation to be super useful here. I think this could be moved to the Methods or SM without great loss.
- Line 251: 'As coal is the cheapest option' Is this true even in 2050 under the high carbon price?
- Figure 4: Good figure and comparisons to help put national results in context.
- Lines 310-313: Probably don't need sentences, 'As this is a minimization problem... battery deployment strategies.'
- Lines 322-323: To what degree is this a function of the net import position on the province? If constraint placed on local supply, would the same finding still hold?
- Figure 7: A helpful comparison, but this could be moved to the SM in order to accommodate some of the other figure suggestions above.
- Line 421: Clearer discussion of follow-on research priorities based on findings. For instance, how could insights change if ancillary services were represented?
- Lines 436-437: Worth sentence or two here on demand profiles.
- Methods: Given the focus on battery storage of the paper, it would be good to list model representation/assumptions of degradation, roundtrip efficiencies, and self-discharge rates.
- Lines 471-474: Province-level variation in labor costs?
- Lines 517-525: It would be good to provide a brief explanation for where the carbon price assumptions come from here (or whether they are simply stylized values).
- Supp. Figure 8: It could be a rendering issue on my machine, but I don't see any data on this curtailment ratio figure.

Reviewer #3 (Remarks to the Author):

Heterogeneous effects of battery storage deployment strategies on the decarbonization of provincial power systems in China

It is an interesting paper that analyses the effects of different battery storage applications across the Chinese provinces. The authors explore the best way to deploy battery storage at the sub-national level. They found that the effects of battery deployment strategies on national electricity transmission and provincial coal-fired power generation vary by province. Moreover, they found that the mix of strategies in intra- and inter-provincial battery deployment results in the lowest system costs. I think that the inclusion of optimized interprovincial transmission power is a nice addition to the article. Also, the fact that the authors explore two scenarios for CO₂ prices is a plus for the article. However, clarity about the electricity mix as well as the type of batteries (and their efficiencies) is missing. Moreover, the claimed differences in future scenarios would require a sensitivity analysis and/or the inclusion of uncertainties in the analysis. I believe that the contribution of the article with respect to the existing literature is rather small and is missing a deeper policy analysis in the discussion. Please find my comments and recommendations below.

- Is the code open-source? Is there any possibility to access it?
- The differences in Fig 1 A, where the authors claim that “Demand-side battery strategy has the largest peak transmission and largest variation, indicating its need for the largest transmission capacity” seem to be small, and given the high uncertainty when looking at prospective scenarios in 2030 and 2050, I believe that the differences are negligible, when these uncertainties are included in the results. The authors could include a sensitivity analysis of their assumed technology costs (that should be available, according to the supplementary material).
- The assumed costs for the different technologies could be shared in the form of a table to analyze the hypotheses of fossil costs and nuclear installations in the future, this is not the case right now.
- Regarding Fig 5, the authors claim that: “Over each time step analyzed, the RE-connected battery strategy decreases coal power generation in the supply province western Inner Mongolia and increases it in a demand province Chongqing, while the Demand-side battery strategy has the opposite effect.” However, when looking at the figure, the same conclusion cannot be reached. The reader sees a decrease in coal power generation in both panels A. Otherwise, if the authors refer to the baseline for 2022, this is not presented. It would be helpful for the reader to see the actual value for the present year which led the authors to conclude that there is an increase in power generation.
- The authors could be more specific and explain within the text why supply provinces can minimize costs by having more RE-connected batteries and fewer Demand-side batteries while demand provinces can minimize costs by having more Demand-side batteries.

- The authors want to assess how to best deploy battery storage at the sub-national level. But here the authors take a top-down vision approach. It would be interesting to discuss how this approach combines or compares with a decentralized bottom-up approach where every actor acts on their own, or how to promote a given behavior within the different provinces.

Clarity and context

- The introduction and/or the discussion should integrate more studies based on China, before prematurely reaching the fact that “The effects of different storage deployment strategies on CO2 emissions at the national/provincial level in China and at multi-year scales is underexplored in the literature.”
- Also, a broader scope of the research on the role of storage linked to CO2 emissions either in the introduction and/or the discussion would be highly appreciated. For instance, discussing the results of the work of Fares and Webber (2017), where residential storage was found to not automatically reduce emissions or energy consumption unless it directly enables renewable energy in Texas.
- When the authors write “These batteries are referred to as ‘Renewable-connected’ batteries since they are charged only from co-located renewables (wind or solar).” in the introduction, do the authors refer to the term “Renewable-connected” as a widely adopted name in the Chinese context or is this for the context of the article? In the literature, the idea of using a battery for different types of uses is referred to as applications (see Battke et al. 2013, Malhotra et al. 2016, Pena-Bello et al. 2019). In this sense, how the authors envisage the provincial deployment of batteries for a single application (e.g. the maximization of renewable energy use or renewable-connected batteries) in the context of the article is not clear to the reviewer.
- What is the role of nuclear in the future Chinese power mix according to the model result? A small comment on the role of the different technologies in the future power mix could be helpful, even if it is not included in the main text.
- The conclusion about “The Mixed battery deployment strategy, in which each province may utilize a mixture of the three deployment strategies, has the lowest total costs of all considered strategies”, is not new, this is a result of benefit stacking, that has been widely studied in the battery storage literature.
- The authors claim a roundtrip energy loss of 12, 15 and 20% depending on the duration used. These values seem very high to the reviewer and looking at the NREL study pointed out by the authors, the values were not found, however, 86% roundtrip efficiency was claimed in one of the referred references from the same webpage. Could the authors be more specific in the citation of the source (https://atb.nrel.gov/electricity/2022/utility-scale_battery_storage), as well as clearly specify the mix of battery technologies used (LiB, Redox...)?
- This is interesting RE-connected and Demand-side battery strategies mainly depend on the characteristics of provincial energy generation and demand. Further recommendations and policy approaches can be given by the authors supported by their results, to give a higher impact to their article.

- The limitations of the study are not addressed within the text.

Suggested improvements

- Please avoid bulk citing references (e.g., "...with energy storage being most promising.2,3,4,5,6,7,8")
- What do the authors mean by "Battery storage has fast response, large energy capacity and can be deployed flexibly" in the introduction?
- What do the authors mean by "decarbonization rates" in section 2.2
- When the authors claim that "Electricity costs decrease rapidly from 2035 to 2050" in section 2.2, it would be helpful to know what the cause of this reduction in the model is. The text seems to imply that it is the deployment of RE, but it is not clear enough. The same applies to the following sentence: "While the Demand-side battery strategy costs are much higher". Also, what are the differences between the scenarios in terms of RE deployment and fossil fuel power plants? This is not clear from the text and does not appear in the tables provided either within the main text or the supplementary material.
- Supplementary Figure 8 does not contain any information
- How Are the annual costs calculated? What is the uncertainty linked to it? What discount rate is used?
- Please use proper mathematical notation with short naming of variables for the equations.
- Figure 4 is not consistent in the display of the calculation of efficiency losses, it should be maintained for both panels.
- Please use the same color code for the figures in the main text as in the supplementary material (see differences e.g. in Fig. 5 and Supplementary Fig. 12).
- In the discussion "When nationally uniform battery deployment takes place, Demand-side and RE-connected batteries have opposite effects on provincial coal-fired power generation and national transmission when they are compared with Grid-connected batteries (Fig. 2)." Should not this refer to Fig 5?
- The sentence "We find that intra- and inter-provincial heterogeneity in battery deployment results in lowest system costs" could be more assertive and easily understandable if modified by something along the lines of "We find that the mix of strategies in intra- and inter-provincial battery deployment results in lowest system costs".
- The authors mix the terms configurations and strategies throughout the text. This should be homogenized to avoid confusion.
- Please include the DOI for the different articles within the references.

General response to Reviewers

We express our sincere appreciation for all reviewers' constructive comments, insightful questions, and specific suggestions. Using their feedback and contributions we have greatly improved our paper. **Our responses (in blue)** together with the reviewers' original comments (in black) and *changes in the manuscript or supplement information (in red italics)* are presented below.

REVIEWER COMMENTS

Reviewer #1 (Remarks to the Author):

This work considers how battery deployment policies/strategies in China can affect grid emissions and transmission needs. The unique contribution is in the nature of the results: by asking a clear modeling question, the authors have addressed an important policy/regulation topic. In my view, the results and framing of the work are well-done and interesting. I do have several concerns and questions about the modeling approach and how it relates to those conclusions, some of which seem important to clearly resolve.

We thank reviewer #1 for their positive comments and questions. Our point-by-point responses are below. We hope our revision and responses address the concerns.

General Comments:

Overall, in these three general comments, I am concerned because I don't understand what is driving the different choices that the model is making. The paper is focused on the outcomes of the model (emissions, transmission builds, etc), but I don't understand why the model is making the choices that it is making – in many cases, I would expect it to make different choices.

1. In section 2 (and several other places): You frequently point out that the RE-sited battery strategy implies the lowest transmission requirement, but I expect that there is an important unstated corollary to this. As far as I can tell, the grid-sited batteries should provide the most benefits to the system, as the RE- and demand-sited scenarios only add constraints to the grid-sited batteries. That is to say: RE- and demand-sited batteries should be strictly inferior from a “battery value” point of view – the grid-sited batteries can do everything that either RE- or demand-sited batteries can do (and more). This seems to be what is described in the methods section. And if that is true, then grid-sited batteries should always be preferable from the point of view of the overall optimization: regardless of the nature of the scenario or objective (lowest cost, lowest emissions, some combination, etc), grid-sited batteries should always be better at achieving it. So while the RE- or demand-sited constraints may reduce

emissions or transmission needs, those are (I suspect) always in the context of grid-sited being the most valuable way to use batteries. So RE-sited batteries may reduce transmission needs more than grid-sited, that is because the additional transmission is of limited cost and decreases the overall system costs in the grid-sited situation. Another way to say it: if limiting transmission was the goal, grid-sited batteries should be no worse than RE-sited batteries (and possibly better) at achieving it.

Overall, I may be misunderstanding the modeling approach or assumptions, but if it is the case that grid-sited batteries are always the best at achieving the objective function (it seems like they should be), then this is a very important thing to discuss, even though it would reduce the implied importance of studying the effects of alternative strategies.

Thank you for your questions. Among the three national strategies, Grid-connected battery strategy is the best in terms of minimizing the system costs, while whether it achieves the lowest CO₂ emissions or not depends on the carbon price. Under low and high carbon prices (See supplementary table 7), RE connected battery strategy achieves the lowest CO₂ emissions. The various battery strategies are valuable to illustrate the trade-offs.

As you mentioned, both RE-connected, and Demand-side battery strategies only impose additional constraints on the Grid-connected battery strategy. When we compare these three strategies from a modelling perspective, the Grid-connected battery strategy is the least restrictive. Any optimal solution obtained using either the RE-connected or Demand-side battery strategy would also be a feasible solution for the Grid-connected battery strategy. Thus, the optimal solution of the Grid-connected battery strategy would result in the lowest objective (total cost) among the optimal solutions of three strategies. This can be confirmed by referring to Supplementary Table 7, which is copied below for your convenience.

Supplementary Table 7 Summary of national results by each battery storage deployment strategy with low carbon prices and rapid battery costs decrease from 2025-2050

	RE-connected batteries	Grid-connected batteries	Demand-side batteries	Mixed batteries
Total cost (Billion dollars)	3118.9	3102.1	3197.6	3075.3
Emission costs	269.6	272.4	292.9	280.1
Gen fixed costs	1563.1	1540.4	1458.9	1494.3
Gen variable costs	86.9	87.2	93.8	88.5
Fuel costs	623.3	616.5	656.1	625.7
Transmission costs	574.7	584.9	593.9	585.6
Others	1.3	1.3	1.2	1.1
Cumulative emissions (Mt)	14104.7	14235.3	15301.4	14544.1
Cumulative renewable capacities (GW)	20921.7	20607.6	17513.3	20357.5
Cumulative battery capacities (GW)	5639.1	6253.7	4808.2	5493

With that said, if limiting transmission expansion is the sole objective, and we set the transmission cost as the objective function, then the Grid-connected battery strategy would achieve the lowest transmission costs out of the three national strategies. However, this analysis would not be useful for policy makers since we cannot ignore the costs of building generators, batteries, and other technologies in reality.

To properly capture the goal of deep decarbonization, we can increase the weight of CO₂ emissions in our objective function. In our model, we investigate this effect by increasing carbon prices. The following table shows the total cost and CO₂ emissions of optimal solutions when we increase the carbon price to 100 times that in the low-carbon-price profile. This price is unrealistic but offers insights on how the system responds to carbon prices. We clearly see that the grid-connected strategy achieves the lowest total cost, as well as CO₂ emissions, among the three strategies. If we compare the mixed strategy with the other three, we can see that the mixed strategy achieves the lowest total cost and CO₂ emissions among the four strategies. (We leave the discussion on the mixed strategy in our response to the next comment from Reviewer #1).

Supplementary Table 1. Summary of national results for each battery storage deployment strategy assuming a 100-fold increase in low carbon prices and rapid battery costs decrease from 2025-2050

	RE-connected batteries	Grid-connected batteries	Demand-side batteries	Mixed batteries
Total cost (Billion dollars)	4088.6	4073.5	4337.1	4064.3
Emission costs	65.2	63.1	112.5	61.1
Gen fixed costs	3072.7	3050.9	3009.0	3049.3
Gen variable costs	41.9	42.0	50.9	42.1
Fuel costs	143.8	135.4	178.7	133.7
Transmission costs	661.2	671.4	885.9	665.0
Others	102.8	110.2	100.0	112.7
Cumulative CO₂ emissions (Mt)	34.3	33.2	59.0	32.1

We have conducted extensive experiments using various carbon prices. We find that the carbon price must be increased to at least 66 times that of the low carbon prices for the Mixed strategy to yield the lowest CO₂ emissions. This suggests that deploying batteries heterogeneously is the best strategy in terms of both minimizing total costs and mitigating CO₂ emissions only if we increase the carbon price as shown in Supplementary table 2. We've added Supplementary Note 6 to explain more about details.

Supplementary Table 2. The threshold of carbon prices (66 times that of the low carbon price scenario in Supplementary table 9) enables the Mixed battery strategy to achieve both the lowest total system costs and CO₂ emissions among the four battery strategies.

	2025	2030	2035	2040	2045	2050
carbon price (RMB/t CO₂)	4950	7920	10890	18480	28380	40260

Supplementary Table 3. Summary of national results for each battery storage deployment strategy assuming a 66-fold increase in low carbon prices and rapid battery costs decrease from 2025-2050

	RE-connected batteries	Grid-connected batteries	Demand-side batteries	Mixed batteries
Total cost (Billion dollars)	4052.1	4039.7	4283.9	4031.3
Emission costs	95.6	91.8	144.3	91.7
Gen fixed costs	3019.6	3010.2	2938.9	3006.7
Gen variable costs	42.4	42.2	52.3	42.1
Fuel costs	196.5	185.9	242.4	182.2
Transmission costs	655.9	664.8	855.8	661.1
Others	42.1	44.8	50.2	47.4
Cumulative CO₂ emissions (Mt)	75.9	72.8	114.7	72.7

2. Another modeling question, possibly related to the one above: if RE- and demand-sited storage are just constrained versions of the grid-sited deployment, then grid-sited should always be preferable (it is strictly more capable). But then why would the “Mixed” strategy be chosen at all? If grid-sited batteries can do everything that RE- and demand-sited ones can do, I would expect that the model should choose 100% grid-sited if offered the option under a “Mixed” strategy. In line 310, you do a good job of explaining why a “mixed” strategy is strictly better than a homogenous strategy – it can do everything that a homogenous strategy can do and more. I would think that this same argument would apply to grid-sited storage relative to the other two types. Line 307 says “The Mixed battery deployment strategy, in which each province may utilize a mixture of the three deployment strategies, has the lowest total costs of all considered strategies.” Why would this be better than 100% grid-sited, if grid-sited batteries provide the most flexibility?

The Mixed strategy is better than 100% grid-connected battery strategy in terms of achieving the lowest total cost (objective) since the Mixed strategy provides the flexibility to align investment decisions with provincial resources.

We understand that our statement might give the reader the impression that the total installed capacity of batteries is fixed, and we are only optimizing the ratio among the three types of batteries. Therefore, we update our statement to the following to make it clearer.

Line 296: “Besides uniform battery deployment strategies, we propose an additional strategy, referred to as the Mixed battery deployment strategy, where each province may utilize any of the three types of battery deployment. The Mixed battery strategy has the lowest total costs

among all considered strategies. This is because unlike the uniform strategies, the Mixed strategy does not constrain all provinces to employ only one type of RE-connected, Grid-connected, or Demand-side batteries.”

In the Mixed strategy setup, all three types of batteries (Grid-connected, RE-connected and Demand-side) may be installed. To explain this implementation, let's consider a simplified example. Suppose we set a decision variable x in the Grid-connected battery strategy, representing the amount of grid-connected battery to be installed in a hypothetical province. Since it's a Grid-connected strategy, there are no additional constraints. To set up the Mixed battery strategy for this province, we introduce three decision variables: x, y and z . Variable x represents the installed capacity of Grid-connected batteries. Variable y represents the installed capacity of RE-connected batteries and we require that these batteries be connected only to RE. Variable z represents the installed capacity of Demand-side batteries which we require be discharged only to the local grid. In this way, the optimization algorithm will solve for the installed capacity of all three types of batteries at minimum cost. However, our model is more complicated than this simplified scenario as we have three different types of batteries with varying durations (1h, 4h and 10h), and the model must allocate each battery type to each 5-year deployment period. Nevertheless, the idea remains the same, and the number of decision variables for installed capacity of batteries in the Mixed strategy is three times as many as in any of the homogeneous strategies. These additional variables allow further optimization by province to minimize costs.

From an optimization point of view, any solution of the grid-connected strategy is feasible for the Mixed strategy. The optimal solution of the Grid-connected battery strategy is no exception. Thus, the optimal solution of the Mixed strategy would achieve a total cost (objective) no greater than the optimal solution of the grid-connected strategy. Our results in Supplementary Table 7 and 8 confirm that this is indeed the case.

From the above analysis, we conclude that among the three homogeneous strategies, the Grid-connected battery strategy is the most flexible one. However, out of the four strategies, the Mixed strategy provides the most flexibility at the provincial level and thus achieves the lowest total cost and results in a solution that is better than grid-connected batteries.

3. This wasn't clearly stated, but I get the impression from the wording that the optimization is done at a province level – is that right? That is, you aren't optimizing for the lowest-cost system for all of China, but rather each province is trying to minimize its internal cost. Whichever way the optimization is designed, it shouldn't affect my comments in #1 and #2 above. But if things are optimized at the province level, how do the provinces choose a Mixed strategy? Wouldn't the optimal strategy in a province depend on the strategy used by the neighbors – a game theory problem? Given that the main arguments in the work are about how different battery strategies encourage or discourage export of electricity, it seems very

relevant to ask what the neighbors are planning, especially if I expect to import or export a lot of power.

We are optimizing at the national level to minimize costs by including a varying carbon price to minimize emissions. The interactions among provinces are accounted for and the strategies of all the provinces are optimized simultaneously. In other words, the model solves for the optimal capacities of generators and installed batteries in each province as well as transmission line capacities among provinces that minimize the national system cost. For further details please see the explanation of the Mixed strategy setup in our previous responses.

We updated our description of the Objective function in the Methods section to read:

Line 465: *“The default objective function in SWITCH-China is the national system cost.”*

Line 473: *“Here C represents the total system costs, E_i represents the total CO₂ emissions in period i , and the weight w_i is the carbon price in period i .”*

Minor Comments:

-I don't think this recent paper is a problem for the work here, but I do think it may make a good comparison point for this analysis, and as a reader of both papers I'm curious if you feel that the results are in agreement or (if not) why the results differ: Mingquan Li, Rui Shan, Edgar Virguez, Dalia Patiño-Echeverri, Shuo Gao, Haichao Ma, Energy storage reduces costs and emissions even without large penetration of renewable energy: The case of China Southern Power Grid, Energy Policy, Volume 161, 2022, 112711, ISSN 0301-4215, <https://doi.org/10.1016/j.enpol.2021.112711>.

Thank you for bringing this paper to our attention. After carefully reading it, we found that it explores the potential benefits of energy storage systems in the context of the China Southern Power Grid. The study analyzes the cost and emission reductions associated with different storage capacity scenarios under various renewable energy penetration levels. The main results of their paper are consistent with our findings. Specifically, their paper mentions that energy storage systems can reduce the cost of electricity generation and emissions, even without a large penetration of renewable energy. In our study, Supplementary Figure 7c-f shows similar results. Battery storage can slightly reduce the CO₂ emissions of the power system even with low renewable penetration in 2025 because it allows full utilization of all renewably generated electricity and RE curtailment. CO₂ emission mitigation will increase with a larger installed capacity of batteries and higher penetration of renewables. Additionally, both this paper and our study show that electricity costs decrease as energy storage capacity increases, which suggests that storage technologies can improve the economics of power systems (Supplementary Figure 7g and 7h).

However, their analysis is on a regional scale, which doesn't include long distance transmission of electricity. Also, their paper didn't consider how various battery deployment strategies could influence the installed capacity of batteries, transmission, and renewable energy, which then further changes the costs and CO₂ emissions of the power system, leaving

a gap in the understanding of the heterogeneous effects of battery storage deployment strategies in China's provincial power system. Our study fills this gap.

We've also added this paper in our literature review in Line 63. Due to space constraints, we discuss this paper in conjunction with other papers that reach similar conclusions.

Reviewer #2 (Remarks to the Author):

This paper uses a power sector capacity expansion model to investigate how the location, scale, and timing of battery storage deployment alters its benefits for various decarbonisation scenarios in China. Although there is a large literature looking at regional storage benefits and value, the paper extends the literature by providing clear scenarios and figures to illustrate alternate locational strategies for batteries across Chinese provinces. The topic and results would be of interest to a range of stakeholders in light of debates about the role of battery storage and decarbonisation strategies in China.

We thank the reviewer#2 for their strong endorsement and positive comments and suggestions. Our point-by-point responses are below.

There are a few issues the authors should revisit to improve the analysis and manuscript. Overall, the manuscript is well written, but there is uneven coverage of scenarios, with some places having too much explanation and others too little. In my view, a missed opportunity in the results section is that the authors do not use figures or text in the body to describe how these alternate strategies alter the generation mix, capacity mix, and configurations of storage. Supplementary Figure 5 does a nice job of showing many of these metrics, and I'd find it valuable to have this in the main text with an accompanying figure that shows variation in capacity by technology (perhaps focusing on a couple particular time periods to show scenario variation). This would help readers to understand implications of these strategies for investments.

Thank you for your suggestions. We've added Supplementary Figure 9 and 10 to show the installed capacity, generation mix and discharged electricity from storage by four battery deployment strategies with low and high carbon prices and rapid decrease battery costs. While we agree that this figure provides valuable information, it is not crucial to our key findings. Due to space limitations we thus prefer to keep the figure in the SI rather than placing it in the body of the manuscript.

*Supplementary Figure 9. Optimal installed capacity of each generation technology from 2025-2050 under RE-connected, Grid-connected, Demand-side, and Mixed battery strategy with **a** Low carbon prices and **b** High carbon prices and rapid decreases in battery costs.*

*Supplementary Figure 10. Optimal power generation of each generator technology and electricity discharged from energy storage from 2025-2050 under RE-connected, Grid-connected, Demand-side, and Mixed battery strategy with **a** Low carbon prices and **b** High carbon prices and rapid decreases in battery costs.*

There were several instances where additional information on the analysis framework and assumptions are important for situating the reader and helping them to better understand the analysis and its potential limitations without necessarily digging through the supplemental material or related papers. SWITCH is a well-established open-source model of capacity expansion, but several areas could be highlighted, in some instances mentioned in the body instead of the Methods section:

- Mention temporal and spatial resolution up front.

Thanks for your suggestion. We've considered putting more details in the framework discussion but feel it would be better to keep the high-level overview of our research questions, methods, and findings. Instead, we specifically refer to the Methods section for details on temporal and spatial resolution and other modeling details.

Line 453: "In our study, we use one hour as a time step and a province as a load zones. To model power system dynamics, four levels of temporal resolution are employed by the SWITCH-China model: 5-year investment periods, months, days, and hours. Our study divides the time span 2025-2050 into six five-year long investment periods: 2023-2028, 2028-2033, 2033-2038, 2038-2043, 2043-2048, 2048-2053. We use 12 months to characterize each investment period, one day to characterizes each month (the median load day) and six hours to characterizes each day. For each day, hourly sampling begins at midnight China Standard Time (CST) and includes the 0th, 4th, 8th, 12th, 16th, and 20th hours. This results in (6 investment periods) \times (12 months/investment period) \times (1 day/month) \times (6 hours/day) = 432 study hours over which the system is dispatched. The results of 2025, 2030, 2035, 2040, 2045 and 2050 are representative years for the six investment periods and are shown in our figures."

- Indicate stringency of power sector policies in terms of levels of price in 2050 and percentage CO₂ reductions. Or something similar to place the stringency of these policies in context.

Our high carbon price scenario with a carbon price of 1220 yuan/t CO₂ in 2050 represents a high level of policy stringency and a strong commitment to achieving carbon neutrality in the power system by around 2050. This scenario is projected to reduce CO₂ emissions by 98% in 2050 compared to 2025, bringing the CO₂ emissions close to carbon neutrality. Our low carbon price scenario, with a carbon price of 610 yuan/t CO₂ in 2050, indicates a moderate level of policy stringency, allowing the power sector to gradually shift from fossil fuels to low carbon power generation. Under this scenario, CO₂ emissions are projected to decrease by 93% from 2025 to 2050, leaving 374Mt of CO₂ emissions in 2050. We've added more elaborations of carbon price in Methods of our manuscript in Line 541.

- Mention choice set of storage options up front. And also that battery duration seems endogenous but only in pre-specified levels of 1, 4, and 10 hours.

Thank you for your suggestion. We add the following sentence to the Methods section around Line 527: *“Regardless of the deployment strategy, we include three battery durations – 1, 4, and 10 hours – to represent various types of battery durations.”*

Battery is typically used for short-term and mid-term storage. We assume battery storage duration can be as long as 10 hours. (BREL, 2021). Thus, in our model, we consider three battery durations – 1, 4, and 10 hours – to represent various types of battery durations.

- Figure 1: Is storage in the ‘RE-connected batteries’ case connected to solar and wind only?

What are the assumptions about cost reductions from coupling? Is ILR endogenous?

Thank you for your questions. Yes, batteries in the “RE-connected batteries” case are connected only to solar and wind. Cost reductions from direct coupling of renewable generation to batteries are uncertain. Therefore, we did not include these changes in our model.

Apologies, but we are not sure what is meant by “ILR”.

- Lines 171-175: How are carbon prices included in the costs?

We are optimizing at the national level to minimize costs by including a varying carbon price to minimize emissions. The total costs of China’s power system includes both system costs and carbon prices. This is explained in Equation (3) (Line 471) and the subsequent paragraph.

- Demand profile source? How much electrification is embedded in these shapes? The authors should consider a sensitivity on load shapes given how previous work indicated that these shapes materially impact storage deployment.

Thank you for your suggestion. In our model, we deploy historical hourly load and generation profiles throughout China. Each date in a future investment period corresponds to an actual date from 2015 for which historical data are available regarding hourly loads, simulated hourly wind and solar capacity factors, and monthly hydroelectric availability. Hourly load data is scaled up to project future demand, while the availabilities of solar, wind, and hydroelectric resources are derived from historical data.

Regarding the effect of electrification of various sectors on demand load curves, we simply applied historical demand profiles from 2015 to future battery deployment and scaled it up using projected future demand.

We’ve added a general description on demand profiles in Line 447: *“Demand profiles in each investment period correspond to 2015 for which historical data is available for hourly loads, simulated hourly wind and solar capacity factors, and monthly hydroelectric availability. Supplementary note 5 shows the Sensitivity analysis of demand loads on battery storage deployment.”*

We've also added a sensitivity analysis of demand loads on battery deployment in Supplementary Note 5.

Supplementary Note 5: Sensitivity analysis of demand loads on battery storage deployment

To evaluate how demand load affects the battery storage deployment, we conduct sensitivity analysis of installed battery capacity under various demand loads. We refer to the projected demand (Supplementary Note 2) and demand profile derived from 2015 as the business as usual (BAU) scenario, then we set up four scenarios - 60% 80%, 120% and 140% of projected demand (60% BAU, 80% BAU, 120% BAU and 140% BAU), to explore the impacts of demand loads on battery storage deployment and the differences in the impacts under four different battery deployment strategies.

We find that, in general, battery deployment has a positive relationship with demand load. Higher demand load results in larger battery storage deployment. However, the correlation between battery storage deployment and demand load is not linear. This is because changes in demand load will further impact the supply and demand balance in each province, which make the situation more complicated. It's valuable to explore this further in future studies.

Supplementary Figure 8 Sensitivity analysis of installed battery capacity from 2025-2050 under BAU, 60%, 80%, 120%, and 140% of BAU demand for (a) RE-connected, (b) Grid-connected, (c) Demand-side, and (d) Mixed battery deployment strategies with low carbon

prices and rapid decreases in battery costs. Numbers in percentage show the ratio of installed battery storage in each demand scenario compared to the BAU demand.

- At the beginning of Section 2.4, I initially missed that these are not only mixed strategies across provinces but also mixed within a province, too. It would be good to make this extra clear.

At the Section 2.4, we have explained the mixed strategy as follows (Line 308) – “each province may utilize a mixture of the three deployment strategies.” We think this should clarify that each province can employ a variety of storage options. We have rephrased the beginning of Section 2.4 as follows.

Line 296: “Besides uniform battery deployment strategies, we propose an additional strategy, referred to as the Mixed battery deployment strategy, where each province may utilize any of the three types of battery deployment. The Mixed battery strategy has the lowest total costs among all considered strategies. This is because unlike the uniform strategies, the Mixed strategy does not constrain all provinces to employ only one type of RE-connected, Grid-connected, or Demand-side batteries.”

Another issue is that the literature review is rather limited. The contributions of this article relate to its geographical coverage and detailed investigation of the value of storage relative to where it is located. However, it would be valuable for readers to see the logic of this contribution expanded while also referencing other prominent studies in the US, EU, and others.

Thank you for your suggestion. We now updated the literature review in our manuscript in Line 49: *“The effects of battery storage on power systems have been explored in many countries⁸⁻¹³, such as the US, EU, Australia, and India. While the benefits of battery storage are clear, there are multiple deployment strategies with complex energy, economic, and emission tradeoffs. Some studies¹⁴⁻¹⁷ highlighted the importance of battery storage deployment strategies and locations in power systems. For example, Schmidt et al¹⁴ found that lifecycle greenhouse gas emissions and costs of storing electricity are determined by battery technology, applications, and geographies. Fares et al¹⁶ discovered that residential energy storage alone does not necessarily lead to reduced emissions if the stored energy comes from non-renewable sources like coal or natural gas. Hittinger et al¹⁷ illustrated that total CO₂ emissions are lower when storage is charged from collocated wind rather than directly from the grid. Craig et al.¹⁸ showed that deploying battery storage would enable a shift of generation from gas- to coal-fired power and therefore increase CO₂ emissions of the power system in Texas. These findings underscore the need to consider deployment strategies and locations to enhance the cost-effectiveness and CO₂ mitigation associated with battery storage in power systems.”*

I had a number of additional suggestions as I reviewed the paper:

- Line 96: Considering using ‘cost-minimizing location, timing, and scale’ instead of ‘best location...’ There are many dimensions to locational choice, so ‘best’ seems too broad and potentially normative.

We’ve revised this sentence in our manuscript.

Line 90: *“These results provide policy makers with the cost-minimizing location, timing, and scale of uniform renewable generators and battery storage installation across China up to 2050.”*

- Figure 1: For Demand-side batteries, I could see where a reader might think these are V2G batteries from EVs based on the picture. Perhaps worth clarifying this in the text and image.

Thanks for pointing out this problem. We’ve removed the Cartoon of EV to avoid this confusion.

- Line 138: ‘Requires’ is too strong, perhaps ‘entails’. And do you mean ‘Large-scale deployment of renewables’? All else equal, storage and transmission are substitutes for each other, as your results demonstrate.

Thanks for your suggestions. We’ve revised this sentence as follows:

Line 156: *“Large-scale deployment of renewables will entail greater electricity transmission capacity.”*

- Figure 2: It would be good to mention scenarios in first sentence of caption instead of at the end. I expected a little more discussion in this section of possible changes in these metrics across the carbon price and storage cost scenarios.

Thank you for your suggestion. We moved the scenarios description to the figure title as follows:

“Fig. 3. Transmitted power and transmission costs of RE-connected, Grid-connected, and Demand-side battery deployment strategies over time with low carbon prices and rapid battery costs decreases.”

- Lines 167-168: It would be helpful to have a more detailed explanation for why this occurs. I think I understand, but it would be valuable for readers to have this spelled out.

Note: Re-connected results in the lowest national CO2 emission

We add the following explanations in Line 137:

“This is because under RE-connected strategy, batteries can only be charged from renewable power generation, resulting in more solar and wind power, and less coal-fired power generation being built to meet demand, which results in the least total CO₂ emissions of any uniform strategy.”

- Lines 173-175: Can this be phrased more directly and quantitatively? Perhaps, ‘Moving from low to high carbon prices increases battery storage from Y to Z GW in 2050, and lower costs increases deployment from Y to Z...’

Thanks for this suggestion. However, in this section, we focus on the trade-offs between national carbon emissions and power system costs under RE-connected, Grid-connected, and Demand-side battery strategies. We’d prefer to present details of how carbon prices influence battery storage in the SI where it appears in SI Notes 3 and 4 because the basic idea of these connections is already understood.

- Lines 224-228: How are system losses accounted for here? I didn’t find this equation to be super useful here. I think this could be moved to the Methods or SM without great loss.

We give two examples to explain how system losses are accounted for.

1. Battery round trip loss: due to losses associated with charging and discharging batteries, battery discharge is always less than the energy used to charge the battery. Thus, the term (battery discharge – battery charge) represents the net battery round trip loss.
2. Transmission loss. Consider a transmission line from province A to province B. There will be energy loss during transmission so that the energy received by province B will be less than the energy transmitted out of province A. The “Net transmitted electricity” of a province is the total energy received from other provinces minus the total energy transmitted out of the province. Here the energy loss during transmission is implicitly considered.

We believe both equations are important here.

Equation 1 is important since it points out that due to curtailment, dispatched electricity is always less than generated electricity. We primarily use dispatched rather than generated electricity in this paper. Thus, clarifying this concept with an equation, we believe is valuable for understanding our results (e.g., Figure 4, and Supplemental Figure 5d, which you suggested we put in the main text).

Equation 2 is important in analyzing and comparing the effect of battery deployment on different provinces. Different deployment strategies result in different dispatched electricity, battery charge/discharge and transmission to meet the demand, which by definition stays the same across different strategies. For example, in around Line 220, we explain the change of battery charge/discharge and net transmission. In order to meet the demand, from Equation 2 we know that the dispatched electricity must increase. As coal is the cheapest option, coal-fired

power generation accounts for 96% of the power generation increase. In the next paragraph around Line 244, we again use Equation 2 to analyze the effect of Demand-side batteries on coal power generation. Moreover, in Figure 4, the black line represents the demand, which stays the same across different strategies. We can also clearly see the energy balance.

Thus, we feel both equations help clarify our results and moving them elsewhere would make the explanation in “Effects of nationally uniform battery deployment strategies” harder.

- Line 251: ‘As coal is the cheapest option’ Is this true even in 2050 under the high carbon price?

Our Figure 4 shows model results in 2040. In 2040, our model results show that coal is the cheapest and most available energy source to provide electricity when the RE-connected battery constraint is added in Chongqing province. To clarify this in our manuscript, we now revised this sentence in Line 235 *“To balance the demand, local power generation increases from 72TWh to 95TWh in 2040. As coal is the cheapest and most available option in Chongqing, coal-fired power generation increases from 32TWh to 54TWh, accounting for 96% of the power generation increase.”*

In the paper we discuss 2040 because this is the year where there are large differences between coal power generation and battery discharge in the provinces we use as examples of supply and demand provinces. By 2050 there is nearly complete decarbonization, so the differences are smaller.

- Figure 4: Good figure and comparisons to help put national results in context.
Thank you! Glad you like it.

- Lines 310-313: Probably don’t need sentences, ‘As this is a minimization problem... battery deployment strategies.’

Thank you for your suggestion. We’ve deleted this sentence in our manuscript.

- Lines 322-323: To what degree is this a function of the net import position on the province? If constraint placed on local supply, would the same finding still hold?

In Lines 322-323, we mentioned “In the nationally uniform strategies Chongqing installs more batteries under the Demand-side deployment strategy than under the Grid- or RE-connected strategies.”

The optimal installed capacity of power generators and batteries in different provinces depends on various factors such as the province's location, available resources, demand and supply requirements, and its interconnection with other provinces through transmission lines. While it

is true that the installed battery capacity of a province under different deployment strategies may be influenced by whether it is a supply or demand province, this relationship is hard to quantify.

- Figure 7: A helpful comparison, but this could be moved to the SM in order to accommodate some of the other figure suggestions above.

Glad you like the comparison. We do think it provides a nice summary of our findings and prefer to keep it in the main manuscript.

- Line 421: Clearer discussion of follow-on research priorities based on findings. For instance, how could insights change if ancillary services were represented?

Thanks for your suggestion. We now added this information in this paragraph as well.

Line 432: “In addition, batteries can offer ancillary services at a lower cost than traditional sources, such as gas-fired peaker plants. By participating in markets for ancillary services, batteries can generate revenue and offset their costs, making them a cost-effective option for providing these services. In future research, deploying a unit commitment power system model would be helpful for exploring these benefits further.”

- Lines 436-437: Worth sentence or two here on demand profiles.

We’ve added a general description on demand profiles in Line 447: *“Demand profiles in each investment period correspond to 2015 for which historical data are available for hourly loads, simulated hourly wind and solar capacity factors, and monthly hydroelectric availability. Supplementary note 5 shows the Sensitivity analysis of demand loads on battery deployment.”*

- Methods: Given the focus on battery storage of the paper, it would be good to list model representation/assumptions of degradation, roundtrip efficiencies, and self-discharge rates.

Thank you for your comments. We’ve added these assumptions of batteries in Line 527:

“Regardless of the deployment strategy, we consider three battery durations – 1, 4, and 10 hours – to represent various types of battery durations. The lifespan of batteries is 15 years. In all our scenarios, we use a roundtrip energy loss from charging/discharging of 15% for 4-h batteries derived from NREL, 2021. For 1-hour and 10-hour batteries, we assume roundtrip loss of 12% and 20%, respectively.”

- Lines 471-474: Province-level variation in labor costs?

We do not include provincial variations in labor costs because they are difficult to project for 2025-2050.

- Lines 517-525: It would be good to provide a brief explanation for where the carbon price assumptions come from here (or whether they are simply stylized values).

Thank you for your suggestion. We included the sources of carbon prices in the Supporting Table S5. We now added this information in this paragraph as well.

Line 544: “The High carbon price scenario represents high level of policy stringency and a strong commitment to achieving carbon neutrality in the power system by around 2050. Low carbon price from 2025 to 2035 is obtained from the projection by the State Grid Energy Research Institution (China Energy & Electricity Outlook 2019), low carbon price after 2035 is based on our own assumptions that power system will not achieve carbon neutrality by 2050 if there are not stringent carbon mitigation policies. High carbon price is set twice the value of low carbon prices from 2025-2050 to represent a more aggressive carbon emission mitigation target.”

- Supp. Figure 8: It could be a rendering issue on my machine, but I don't see any data on this curtailment ratio figure.

Thank you for pointing out this issue. It shows well in our screens. We've updated this figure and hopefully you are able to see the data now.

Reviewer #3 (Remarks to the Author):

Heterogeneous effects of battery storage deployment strategies on the decarbonization of provincial power systems in China

It is an interesting paper that analyses the effects of different battery storage applications across the Chinese provinces. The authors explore the best way to deploy battery storage at the sub-national level. They found that the effects of battery deployment strategies on national electricity transmission and provincial coal-fired power generation vary by province. Moreover, they found that the mix of strategies in intra- and inter-provincial battery deployment results in the lowest system costs. I think that the inclusion of optimized interprovincial transmission power is a nice addition to the article. Also, the fact that the authors explore two scenarios for CO₂ prices is a plus for the article. However, clarity about the electricity mix as well as the type of batteries (and their efficiencies) is missing. Moreover, the claimed differences in future scenarios would require a sensitivity analysis and/or the inclusion of uncertainties in the analysis. I believe that the contribution of the article with respect to the existing literature is rather small and is missing a deeper policy analysis in the discussion. Please find my comments and recommendations below.

We appreciate your concerns about the clarity and deeper policy analysis in our study. We improved the manuscript with the help of your suggestions. Our point-by-point responses are below.

- Is the code open-source? Is there any possibility to access it?

Yes, our codes are open source. You can find the key codes of SWITCH-China at GitHub with this link: <https://github.com/switch-model>. Other codes for results analysis and plots will be published in an archive and put the link in our manuscript when we finalize all revisions. We have updated our description in the Data availability and Code availability in our manuscript.

- The differences in Fig 1 A, where the authors claim that “Demand-side battery strategy has the largest peak transmission and largest variation, indicating its need for the largest transmission capacity” seem to be small, and given the high uncertainty when looking at prospective scenarios in 2030 and 2050, I believe that the differences are negligible, when these uncertainties are included in the results. The authors could include a sensitivity analysis of their assumed technology costs (that should be available, according to the supplementary material).

In this paragraph, the purpose of Fig. 3b is to shed light on the following questions – Why do Demand-side batteries require the highest transmitted capacity in 2050 (Fig. 3a) but actually transmit less electricity than Grid-connected batteries in 2050 (Fig. 3c)? We appreciate your pointing out the lack of clarity in our earlier description and rephrase the whole paragraph below:

“The Demand-side battery strategy requires the highest transmission capacity (Fig. 3a) which results in the highest transmission costs (Fig. 3d). However, this strategy actually transmits less electricity than the Grid-connected battery scenario in 2045 and 2050 (Fig. 3c). This can be explained by Fig. 3b - transmission power profiles of the three strategies in an example two-day period in 2050. The peak of the transmission power profile indicates transmission capacity requirements, where the Demand-side strategy is slightly larger than other strategies. The areas under the curves represent transmitted electricity, where Demand-side strategy is less than the Grid-connected battery, even though its peak transmission is higher. This explains the changing magnitudes of the three strategies in 2040 and 2045-2050 in Fig. 3a and Fig. 3c.”

We’ve also added a sensitivity analysis about the effects of various technology costs on installed capacity, transmitted power, and annual transmission costs and we’ve mentioned it in the main manuscript, which refer the interested reader to the SI figures for details.

Line 172: *“We also conduct a sensitivity analysis about the effects of various technology costs on installed capacity, transmitted power, and annual transmission costs in Supplementary Figure 20-22. We find that more rapid decrease of technology costs results in larger differences among the three national battery strategies.”*

Supplementary Figure 20. Transmission capacity of high and low carbon prices with **a** base case of battery and renewable costs, **b** moderate decrease of battery and renewable costs, and **c** rapid decrease of battery and renewable costs.

Supplementary Figure 21. Transmitted power of high and low carbon prices with **a** base case of battery and renewable costs, **b** moderate decrease of battery and renewable costs, and **c** rapid decrease of battery and renewable costs.

Supplementary Figure 22. Annual transmission costs of high and low carbon prices with **a** base case of battery and renewable costs, **b** moderate decrease of battery and renewable costs, and **c** rapid decrease of battery and renewable costs.

- The assumed costs for the different technologies could be shared in the form of a table to analyze the hypotheses of fossil costs and nuclear installations in the future, this is not the case right now.

We have provided our capital cost assumptions for batteries, solar PV, onshore wind, and offshore wind from 2020-2050 in Supplementary Figures 2-4. Additionally, we have included the costs of fossil fuel and nuclear power generators in Supplementary Table 4.

Supplementary Table 4. National average capital cost, fixed operation and maintenance costs and fuel costs of coal-fired, natural gas and nuclear power plants from 2020-2050 (BNEF, 2022; He et al., 2016).

Costs	Year	Coal	Gas	Nuclear
Capital costs (\$/KW)	2020	598	532	2140
	2030	598	532	2140
	2040	598	532	2140
	2050	598	532	2140
Fixed O&M (\$/kW)	2020	10.6	13.2	72
	2030	10.6	13.2	72
	2040	10.6	13.2	72
	2050	10.6	13.2	72
Fuel costs (\$/MMBtu)	2020	4.5	13.7	0.82
	2030	4.8	15.6	0.90
	2040	5.3	16.9	1.06
	2050	5.8	16.9	1.22

- Regarding Fig 5, the authors claim that: “Over each time step analyzed, the RE-connected battery strategy decreases coal power generation in the supply province western Inner Mongolia and increases it in a demand province Chongqing, while the Demand-side battery strategy has the opposite effect.” However, when looking at the figure, the same conclusion cannot be reached. The reader sees a decrease in coal power generation in both panels A. Otherwise, if the authors refer to the baseline for 2022, this is not presented. It would be helpful for the reader to see the actual value for the present year which led the authors to conclude that there is an increase in power generation.

Thank you for pointing out this concern. The conclusion is summarized by comparing RE-connected battery strategy and Demand-side battery strategy with Grid-connected battery strategy. We’ve clarified the sentence as follows:

Line 274: “Over each time step analyzed, compared with the Grid-connected battery strategy, the RE-connected battery strategy decreases coal power generation in the supply province

(western Inner Mongolia) and increases it in the demand province (Chongqing), while the Demand-side battery strategy has the opposite effect.”

Also, we now updated Figure 5 by adding the actual number of coal-fired power generation and discharged electricity from batteries.

Figure 5. Coal-fired power generation and power discharged from batteries in a typical supply and demand province. Comparison of a coal-fired power generation and b power discharged from batteries in a supply province (western Inner Mongolia) and a demand province (Chongqing) for three battery storage deployment strategies with low carbon prices and rapid battery cost decreases. Note the very different scales for Chongqing and Western Inner Mongolia which occur because Chongqing is a typical city with power demand that

exceeds supply with little renewable resources while western Inner Mongolia is a province with a demand far smaller than its power supply and a large supply of renewable resources. Effects of other scenarios are shown in the Supplementary Fig.15-18. Numbers in red, yellow, and blue represent the actual value of coal-fired power generation and discharged electricity from batteries under RE-connected, Grid-connected, and Demand-side battery strategies.

- The authors could be more specific and explain within the text why supply provinces can minimize costs by having more RE-connected batteries and fewer Demand-side batteries while demand provinces can minimize costs by having more Demand-side batteries.

Since we are minimizing the national system cost as a whole, each province cannot minimize its own cost individually. Thus, we interpret the reviewer's question as "In the mixed strategy, which achieves the lowest national system cost, why do supply provinces have more RE-connected batteries than Demand-side batteries while demand provinces have more Demand-side batteries than RE connected batteries in Figure 6?"

We've added our response to this question in Line 311: *"This is because, for demand provinces, their local power generation is less than the total demand, and they usually have limited renewable resources. Demand-side batteries can store electricity transmitted from supply provinces during the day and meet the high demand during the evening."*

Line 319: *"The reason is, for supply provinces, their total local power generation exceeds their total demand through 2050 and these supply provinces are mostly in western China with low population density and have abundant renewable resources. RE-connected batteries in supply provinces can store excess renewable electricity and transmit it to demand provinces to meet their electricity needs. Therefore, in the mixed strategy, supply provinces choose to install more RE-connected batteries than Demand-side batteries, while demand provinces make the opposite choice."*

- The authors want to assess how to best deploy battery storage at the sub-national level. But here the authors take a top-down vision approach. It would be interesting to discuss how this approach combines or compares with a decentralized bottom-up approach where every actor acts on their own, or how to promote a given behavior within the different provinces.

Thanks for your question – it's interesting, but beyond the scope of our paper.

From a game-theoretic perspective, the bottom-up approach may not lead to the global minimum because each province pursuing its own self-interest can result in suboptimal outcomes for the nation as a whole.

In the bottom-up approach, there may be a temptation for each province to free-ride on the efforts of others. For example, if one province invests heavily in renewable energy sources,

other provinces may choose not to invest as much, assuming they can benefit from the clean energy produced by the first province without bearing the same costs.

On the other hand, a top-down approach that minimizes the total cost for all provinces takes into account the collective interests of all provinces, rather than just the interests of individual actors. This approach encourages provinces to work together towards a common goal, and to coordinate their actions in a way that maximizes overall efficiency.

By adopting a top-down approach, we provide a national-level solution that can be utilized by policymakers to encourage specific behaviors in specific provinces with maximum effectiveness.

Clarity and context

- The introduction and/or the discussion should integrate more studies based on China, before prematurely reaching the fact that “The effects of different storage deployment strategies on CO₂ emissions at the national/provincial level in China and at multi-year scales is underexplored in the literature.”

Thank you for your suggestion. We have updated the literature review before this sentence to better support our assertion.

Line 80: “Other studies¹⁹⁻²² focus on the role of battery storage deployment in China’s power system. For example, He et al.⁵ and Liu et al.²²’s research suggested that the deployment of energy storage systems can help reduce carbon emissions by facilitating renewable energy integration and improving the overall efficiency of the power system. However, the effects of different storage deployment strategies on total system costs and CO₂ emissions at the national/provincial level in China and at multi-year scales is underexplored in this literature.”

- Also, a broader scope of the research on the role of storage linked to CO₂ emissions either in the introduction and/or the discussion would be highly appreciated. For instance, discussing the results of the work of Fares and Webber (2017), where residential storage was found to not automatically reduce emissions or energy consumption unless it directly enables renewable energy in Texas.

Thank you for bringing this paper to our attention. After carefully reading it, we found that Fares and Webber (2017) emphasized the need to integrate residential energy storage with renewable energy resources in order to achieve their full potential in reducing carbon emissions and energy sources. They discovered that residential energy storage alone does not necessarily lead to reduced emissions. The emissions associated with stored energy depend on the source of that energy. If the stored energy comes from non-renewable sources like coal or natural gas, the emissions will not be reduced. To effectively reduce emissions, residential energy storage

systems should be used in conjunction with renewable energy sources. This is consistent with our findings. However, when we consider all provinces in China, the situation is more complicated. That's the reason we established a mixed battery deployment strategy to capture the heterogenous effects of battery storage in both the national and provincial power systems.

We now have added a broader discussion on the role of storage in CO₂ emissions of the power system including the paper you mentioned in our Introduction:

Line 49: *“The effects of battery storage on power systems have been explored in many countries⁸⁻¹³, such as the US, EU, Australia, and India. While the benefits of battery storage are clear, deployment strategies involve complex energy, economic, and emission tradeoffs. Some studies¹⁴⁻¹⁷ highlight the importance of battery storage deployment strategies and their location in power systems. For example, Schmidt et al¹⁴ found that lifecycle greenhouse gas emissions and costs of storing electricity are determined by battery technology, applications, and geographies. Fares et al¹⁶ discovered that residential energy storage alone does not necessarily lead to reduced emissions if the stored energy comes from non-renewable sources like coal or natural gas. Hittinger et al¹⁷ illustrated that total CO₂ emissions are lower when storage is charged from collocated wind rather than directly from the grid. Craig et al.¹⁸ showed that deploying battery storage would enable a shift of generation from gas- to coal-fired power and therefore increase CO₂ emissions of the power system in Texas. These findings underscore the need to consider deployment strategies and locations to enhance the cost-effectiveness and CO₂ mitigation associated with battery storage in power systems.”*

- When the authors write “These batteries are referred to as ‘Renewable-connected’ batteries since they are charged only from co-located renewables (wind or solar).” in the introduction, do the authors refer to the term “Renewable-connected” as a widely adopted name in the Chinese context or is this for the context of the article? In the literature, the idea of using a battery for different types of uses is referred to as applications (see Battke et al. 2013, Malhotra et al. 2016, Pena-Bello et al. 2019). In this sense, how the authors envisage the provincial deployment of batteries for a single application (e.g. the maximization of renewable energy use or renewable-connected batteries) in the context of the article is not clear to the reviewer.

The term "Renewable-connected" is not a widely adopted name in the Chinese context. This name is inspired by the "centralized renewable energy + energy storage" phrase, which is directly translated from the Chinese incentive policies. We created this concise term for the context of the article. In our manuscript, we gave more details of the implications of battery deployment strategies in the “Battery deployment strategy” section of “Scenario design” in the Methods (Line 499- 565).

Motivated by these incentive policies, we investigate how "RE-connected," "Grid-connected," and "Demand-side" battery deployment strategies affect China's carbon neutrality goal pathway. These deployment strategies are not tied to specific applications. For instance, both Grid-connected and Demand-side batteries can be used for backup power or energy arbitrage, while both RE-connected and Grid-connected batteries can be utilized for peak shaving and frequency regulation. Therefore, we opted for non-application-specific names to better reflect our intent and connect back to the incentive policies.

Since the article's goal is not to advocate for any single application, we did not analyze how these applications would benefit from the deployment strategies. Instead, our results could help policymakers deploy battery storage strategies at the provincial level for the purpose of decarbonization and cost minimization.

- What is the role of nuclear in the future Chinese power mix according to the model result? A small comment on the role of the different technologies in the future power mix could be helpful, even if it is not included in the main text.

Thank you for suggesting this. We now added the paragraph below to elaborate on the role of nuclear in China's future power system in Supplementary Note 3.

“In addition, we find that nuclear power plays an important role in China's power system, especially when the penetration of renewables is high. Nuclear generators produce a consistent and reliable output, offering baseload power to stabilize the grid and along with batteries mitigate the intermittency of renewable energy sources.”

- The conclusion about “The Mixed battery deployment strategy, in which each province may utilize a mixture of the three deployment strategies, has the lowest total costs of all considered strategies”, is not new, this is a result of benefit stacking, that has been widely studied in the battery storage literature.

We've added these conclusions below in our manuscript in Line 362:

“Our work is the first, of which we are aware, that analyzes battery storage options for each province individually before optimizing for minimum cost at the national level. Our disaggregated work (that allows differentiation between provinces rich in coal and rich in renewable generation) not only confirms early national findings but also provides a higher resolution for the variability existing among provinces.”

- The authors claim a roundtrip energy loss of 12, 15 and 20% depending on the duration used. These values seem very high to the reviewer and looking at the NREL study pointed out by the authors, the values were not found, however, 86% roundtrip efficiency was claimed in one of the referred references from the same webpage. Could the authors be more specific in the citation of the source (<https://atb.nrel.gov/electricity/2022/utility->

scale battery storage), as well as clearly specify the mix of battery technologies used (LiB, Redox...)?

In the “Utility-Scale battery storage” sheet of the file named “2021 Corrected Annual Technology Baseline Workbook for MAC from 8-12-2021” found on the website: <https://data.openei.org/submissions/4129>, the average roundtrip energy loss is 15%. This is very close to the 14% loss in the NREL study the reviewer mentions. We assume that longer duration batteries may experience higher energy losses over time due to factors such as self-discharge and degradation, which could impact their overall performance. Thus, we use 15% roundtrip energy loss for 4-hour batteries, 12% for 1-hour batteries and 20% for 10-hour batteries. Since we apply these assumptions for all scenarios, we think they will not significantly influence our final results. Additionally, the data mentions that the representative battery is Li-ion battery storage, which is also the representative battery in our study.

We now clarified this in our manuscript in line 529:

“In all our scenarios, we use a roundtrip energy loss from charging/discharging of 15% for 4-h batteries derived from NREL, 2021. For 1-hour and 10-hour batteries, we assume roundtrip loss of 12% and 20%, respectively.”

- This is interesting RE-connected and Demand-side battery strategies mainly depend on the characteristics of provincial energy generation and demand. Further recommendations and policy approaches can be given by the authors supported by their results, to give a higher impact to their article.

Thank you for your suggestion. We have now added additional recommendations and policy implications to our manuscript in Line 414.

“We find that strategic deployment of battery storage will influence both system costs and carbon emissions from China’s power system. To further accelerate the transition to a carbon neutral power system, targeted government programs which provide financial incentives to encourage the adoption of RE-connected batteries in supply provinces and Demand-side connected batteries in demand provinces would facilitate more rapid power system decarbonization. In addition, improving transmission capacity and efficiency to facilitate the exchange of renewable energy between supply and demand provinces could maximize the benefits of battery storage deployment strategies.”

- The limitations of the study are not addressed within the text.

We have discussed limitations of our study in the last paragraph of the Discussion section.

We now add additional limitations as shown below.

Line 430: *“We use Li-ion batteries to represent all batteries in our study. However, as battery storage technology advances, alternatives should be assessed and incorporated into future*

battery storage deployment policies. In addition, batteries can offer ancillary services at a lower cost than traditional sources, such as gas-fired peaker plants. By participating in markets for ancillary services, batteries can generate revenue and offset their costs, making them a cost-effective option for providing these services. In future research, deploying a unit commitment power system model would be helpful for exploring these benefits further.”

Suggested improvements

- Please avoid bulk citing references (e.g., “...with energy storage being most promising.^{2,3,4,5,6,7,8}”)

OK. Thank you for this point. We revised the format for citing references.

Line 42: *“Various technologies can smooth this variability, with energy storage being most promising²⁻⁸.”*

- What do the authors mean by “Battery storage has fast response, large energy capacity and can be deployed flexibly” in the introduction?

Thank you for pointing out the lack of clarity of this sentence. Fast response refers to battery storage systems’ ability to quickly respond to fluctuations in energy demand or supply. They can rapidly discharge stored energy when needed or charge when excess energy is available.

Large energy capacity implies that batteries can store a significant amount of energy, making them suitable for various applications, from residential to grid-scale energy storage. This capacity allows batteries to store excess energy during periods of low demand and release it during peak demand, reducing the need for conventional, emission-intensive peaker plants.

Flexible deployment indicates that battery storage systems can be installed and utilized in various locations and configurations, ranging from small residential setups to large-scale installations connected to the grid.

We now revised this sentence as follows:

“Battery storage allows rapid energy discharges to smooth fluctuations in electricity supply. It also offers substantial storage capacity and can be deployed in various locations and strategies.”

- What do the authors mean by “decarbonization rates” in section 2.2

Decarbonization rates refer to the rate at which a country, company, or other entity reduces the amount of carbon emissions it produces over a given period of time. Here, it specifically refers to the rate of CO₂ emission mitigation in the power system. To avoid confusion, we’ve deleted the term in the sentence.

- When the authors claim that “Electricity costs decrease rapidly from 2035 to 2050” in section 2.2, it would be helpful to know what the cause of this reduction in the model is. The text seems to imply that it is the deployment of RE, but it is not clear enough. The same applies to the following sentence: “While the Demand-side battery strategy costs are much higher”. Also, what are the differences between the scenarios in terms of RE deployment and fossil fuel power plants? This is not clear from the text and does not appear in the tables provided either within the main text or the supplementary material.

Thank you for pointing out the need to clarify our results. We’ve added an explanation of this statement in Line 134: “*..with electricity costs decreasing rapidly from 2035 to 2050 due to the decreasing renewable generation costs during this period.*”

For the electricity cost comparison, we’ve provided total system costs under RE-connected, Grid-connected, and Demand-side battery strategies in Supplementary Table 5 and 6. We’ve added Supplementary Figure 9 and 10 to show the installed capacity and power generation of renewables and fossil fuel power generators under various battery strategies.

- Supplementary Figure 8 does not contain any information
Supplementary Figure 11 (updated from 8 to 11) includes the annual average curtailment ratio of renewables for each province under eight scenarios. The original figure has a rendering issue. We’ve updated the figure in our supplementary file and hope you are able to see the data now.
- How Are the annual costs calculated? What is the uncertainty linked to it? What discount rate is used?

In our model, we use capital costs, fixed O&M and fuel costs as input, with more details available in the Objective function section of Methods. We apply an 8% discount rate based on Gang et al (2021). The SWITCH model employs net present value (NPV) as the key financial parameter in the objective function. More details can be found in 4.3.2 Financials in the Supplementary material of Johnston et al., 2019 (<https://ars.els-cdn.com/content/image/1-s2.0-S2352711018301547-mmc1.pdf>).

The uncertainties in these calculations come from capital costs, O&M costs, and the discount rate. We believe that the capital cost is the most significant uncertainty factor for our battery deployment as it is difficult to predict future price changes. Therefore, we’ve conducted a sensitivity analysis of capital costs for various renewable energy and batteries in our results, as shown in Supplementary Figure 20-22.

- Please use proper mathematical notation with short naming of variables for the equations. Thanks for this suggestion. For the equation in Line 471, we have changed it as follows:

Minimize: $C + \sum_{i \in p} w_i \mathcal{E}_i$

Where C represents the total system costs, \mathcal{E}_i represents the total CO₂ emissions in period i , and the weight w_i is the carbon price in period i .

For the balancing equation in Line 226, we can modify it as follows:

$$\mathcal{R} = \mathcal{D}\mathcal{E} + (\mathcal{B}_c - \mathcal{B}_d) + \mathcal{T}_{net}$$

However, since these notations would only appear once here, we believe that using descriptive names instead of symbols in the equation might be easier for readers. Thus, we are inclined to keep Lines 208 and 210 as they are. If the reviewer or the editor thinks it is necessary to use symbol notations, we will update the equations after receiving feedback.

- Figure 4 is not consistent in the display of the calculation of efficiency losses, it should be maintained for both panels.

Figure 4 includes the efficiency losses in both panels. In both panels, light purple represents electricity being charged to batteries, and deep purple represents electricity discharged from batteries. The absolute value of light purple is 12-15% higher than the value of deep purple, representing the efficiency loss of batteries.

- Please use the same color code for the figures in the main text as in the supplementary material (see differences e.g. in Fig. 5 and Supplementary Fig. 12).

Thank you for your suggestion! We now use the same color code for the figures in the main text and in the supplementary material.

- In the discussion “When nationally uniform battery deployment takes place, Demand-side and RE-connected batteries have opposite effects on provincial coal-fired power generation and national transmission when they are compared with Grid-connected batteries (Fig. 2).” Should not this refer to Fig 5?

Thanks for pointing this out. We agree that this sentence should refer to both Figure 2 and Figure 5 since it mentions both provincial coal-fired power generation and national transmission. We’ve added Fig. 5 in the brackets.

- The sentence “We find that intra- and inter-provincial heterogeneity in battery deployment results in lowest system costs” could be more assertive and easily understandable if modified by something along the lines of “We find that the mix of strategies in intra- and inter-provincial battery deployment results in lowest system costs”.

Thank you for your suggestion. We’ve revised this sentence based on your suggestion.

- The authors mix the terms configurations and strategies throughout the text. This should be homogenized to avoid confusion.

Thanks for pointing out this. We've replaced the "configurations" with "strategies" throughout the manuscript.

- Please include the DOI for the different articles within the references.

Thank you for your suggestions. The current format is based on Nature Communication's formatting requirements, which don't include the DOI.

REVIEWERS' COMMENTS

Reviewer #1 (Remarks to the Author):

I'm happy with the changes made in response to reviewer comments. I find the work to be interesting and relevant.

Reviewer #2 (Remarks to the Author):

I am satisfied with the authors' revisions and recommend publication.

Reviewer #3 (Remarks to the Author):

I am pleased to express my satisfaction with the amendments made by the article's authors in response to the comments I provided. The authors have shown a commendable dedication to addressing the concerns and suggestions raised, resulting in a significantly improved manuscript. Their thoroughness in the revision has considerably strengthened the overall quality of the article. I believe the revised article is now well-positioned to make a valuable contribution to the field.

I only have two minor additional comments, first regarding Eq. 1 and 2 I would prefer a shorter notation as in the Methods, but it is up to the authors and the editor. Second, Ref. [1] says Chian instead of China.

General response to Reviewers

We express our sincere appreciation for all reviewers' constructive comments. Our responses (in blue) together with the reviewers' original comments (in black) and changes in the manuscript or supplementary information (in red) are presented below.

Reviewer #1 (Remarks to the Author):

I'm happy with the changes made in response to reviewer comments. I find the work to be interesting and relevant.

Reponses: We sincerely appreciate your positive feedback and the time you've dedicated to review our work. We are glad to know that the changes we made in response to your comments were satisfactory.

Reviewer #2 (Remarks to the Author):

I am satisfied with the authors' revisions and recommend publication.

Reponses: We truly appreciate your thoughtful review and the time spent on our manuscript. Thank you for your recommendation for publication of our work. Your input has played a crucial role in its' improvement.

Reviewer #3 (Remarks to the Author):

I am pleased to express my satisfaction with the amendments made by the article's authors in response to the comments I provided. The authors have shown a commendable dedication to addressing the concerns and suggestions raised, resulting in a significantly improved manuscript. Their thoroughness in the revision has considerably strengthened the overall quality of the article. I believe the revised article is now well-positioned to make a valuable contribution to the field.

I only have two minor additional comments, first regarding Eq. 1 and 2 I would prefer a shorter notation as in the Methods, but it is up to the authors and the editor. Second, Ref. [1] says Chian instead of China.

Responses: We greatly appreciate your encouraging feedback and acknowledgment of the efforts we've put into revising our manuscript.

We address your two minor comments below. First, regarding the notation in Eq. 1 and 2, we've updated them as shown in equation (1) as follows:

$$\mathcal{D} = \mathcal{DE} + (\mathcal{B}_a - \mathcal{B}_c) + \mathcal{T}_{net}$$

(1)

The equation shows the energy balance of each province, which is the key to explaining the

effects of battery deployment strategies on power generation and CO₂ emissions. Here, \mathcal{D} represents the provincial demand, which equals the sum of three terms - locally dispatched electricity \mathcal{DE} , local net battery discharge (the difference of battery discharge \mathcal{B}_d and battery charge \mathcal{B}_c), as well as the net transmitted electricity into the province \mathcal{T}_{net} . The dispatched electricity \mathcal{DE} equals generated electricity minus curtailed electricity.

Second, we apologize for the typo in Ref. [1]. It will be corrected to "China" immediately.